# Dissociation Mode of the O–H Bond in Betanidin, pK_a_-Clusterization Prediction, and Molecular Interactions via Shape Theory and DFT Methods

**DOI:** 10.3390/ijms24032923

**Published:** 2023-02-02

**Authors:** Iliana María Ramírez-Velásquez, Álvaro H. Bedoya-Calle, Ederley Vélez, Francisco J. Caro-Lopera

**Affiliations:** 1Faculty of Exact and Applied Sciences, Instituto Tecnológico Metropolitano, Medellín 050034, Colombia; 2Faculty of Basic Sciences, University of Medellin, Medellín 050026, Colombia

**Keywords:** betanidin, linear model, cluster prediction, pK_a_ prediction, Riemann–Mulliken distance, deprotonation, shape theory

## Abstract

Betanidin (Bd) is a nitrogenous metabolite with significant bioactive potential influenced by pH. Its free radical scavenging activity and deprotonation pathway are crucial to studying its physicochemical properties. Motivated by the published discrepancies about the best deprotonation routes in Bd, this work explores all possible pathways for proton extractions on that molecule, by using the direct approach method based on pK_a_. The complete space of exploration is supported by a linear relation with constant slope, where the pK_a_ is written in terms of the associated deprotonated molecule energy. The deprotonation rounds 1, …, 6 define groups of parallel linear models with constant slope. The intercepts of the models just depend on the protonated energy for each round, and then the pK_a_ can be trivially ordered and explained by the energy. We use the direct approximation method to obtain the value of pK_a_. We predict all possible outcomes based on a linear model of the energy and some related verified assumptions. We also include a new measure of similarity or dissimilarity between the protonated and deprotonated molecules, via a geometric–chemical descriptor called the Riemann–Mulliken distance (RMD). The RMD considers the cartesian coordinates of the atoms, the atomic mass, and the Mulliken charges. After exploring the complete set of permutations, we show that the successive deprotonation process does not inherit the local energy minimum and that the commutativity of the paths does not hold either. The resulting clusterization of pK_a_ can be explained by the local acid and basic groups of the BD, and the successive deprotonation can be predicted by using the chemical explained linear models, which can avoid unnecessary optimizations. Another part of the research uses our own algorithm based on shape theory to determine the protein’s active site automatically, and molecular dynamics confirmed the results of the molecular docking of Bd in protonated and anionic form with the enzyme aldose reductase (AR). Also, we calculate the descriptors associated with the SET and SPLET mechanisms.

## 1. Introduction

Betalains are compounds that, in their basic structure, contain nitrogen. In this way, they are also identified as chromoalkaloids and characterized by being soluble in water. These compounds are present in a restricted number of plants of the order *Caryophyllales* and some of the genus *Basidiomycetes*. For instance, in the genus *Amaran-thaceae*, betalains can be found in leaves as well as flowers; in the genera *Hylocereus*, *Stenocereus*, and *Opuntia*, it can be found in the fruits, and in the genera *Beta* and *Rheum* in the roots and stems, respectively [1,2,3]. Moreover, betalains have a wide range of colors like yellow-orange for betaxantins and red-purple for betacyanins. This situation gives them the potential to be used as natural colorants [4]. Further, betalains can provide functional properties to the food, producing a beneficial effect on health [5,6]. In pharmaceutical and cosmetic applications, betalains are stable in a pH range of 3.5 to 7. This characteristic gives them value for being used in low acid and neutral products [6,7]. The betacyanins are immonium conjugates of betalamic acid with 3,4-dihydroxyphenylalanine (cyclo-DOPA), forming betanidin, which contains phenolic and cyclic amine groups, both of which are very good electron donors, acting as antioxidants [8]. Hydrolysis of betacyanin produces betanidin, and their conjugated double bond is associated with their color. The maximum light absorption at 540 nm characterizes the red betacyanins. Betanin (Bn) is composed of betanidin (Bd) and linked β-glycosidic with glucose at C5 [9,10], as can be seen in Figure 1.

We present several studies that report the efficacy of betalains for preventing the oxidation of biological molecules induced by active oxygen [8,11]. In addition, other studies report that the antioxidant capacity of Bn is pH-dependent by TEAC (trolox equivalent antioxidant capacity) assay; this situation was analyzed from the calculation of the phenolic OH homolytic bond dissociation energy (BDE) and the ionization potential (IP). These parameters were obtained via DFT B3LYP/6-311+G** or B3LYP/6-31G** quantum-mechanical calculations [10]. Other DFT studies report the reactivity in six pathways against some ordinary radicals, such as hydroxyl, hydroperoxide, superoxide, and nitric oxide [12].

On the other hand, studies via DFT computations show the role of the hydroxyl group position in the radical scavenging and antioxidant activity of some compounds to delineate structure–reactivity patterns and therefore the most probable mechanism of interaction between these compounds and free radicals. In this case, the most probable mechanism in water is SPLET (sequential proton-loss electron transfer mechanism) [13].

The thermochemical parameter describing the hydrogen and electron donation ability by betanin is OH bond dissociation energy (BDE), representing the ease of hydrogen atom donation. Each mono-deprotonated form of betanin is a better H donor (lower BDE values) than betanin in its cationic form [10]. However, betalains are rich in electron density, which suggests that the mechanisms related to parameters representing electron donation should be responsible for free radical-scavenging activity [14,15]. The bioactive potential of betalains is influenced by factors such as pH, temperature, light, water activity, oxygen, metal ions, and enzymatic action [16].

The pK_a_ value is an important parameter to elucidate the oxidation mechanism of any substance. Bd belongs to the general group of betalains. The experimental pK_a_ values of Bd were limited to a single deprotonation step. In this work, we have used the density functional theory (DFT) and the density-based solvation model (SMD) to calculate the corresponding pK_a_ values. However, there are several methodologies for computing the pK_a_; see, for example, [17,18,19].

Based on the information presented and the importance of betalains, we consider that studies on the activity of these compounds are not yet sufficient. Therefore, the present work aims to expand the research in this line by approaching the modeling of betanidin (Bd) through DFT methods in order to establish the molecular descriptors associated with the free radical scavenging activity and the deprotonation pathway of the molecule in question. First, we used the direct approach method to calculate the pK_a_ values of Bd [20]. Then, we explored the potential energy surface defined by all possible mono-deprotonations (at the three carboxylic positions, the two hydroxyl groups, and N16) and the subsequent deprotonations of Bd. Finally, we analyzed all three possible deprotonation routes of Bd (1st: 6 paths, 2nd: 30 paths, 3rd: 120 paths) by implementing an invariant measure based on shape theory, including atomic mass and Mulliken charges. The measurement of the similarity or dissimilarity arises from shape theory using the concept of Riemann–Mulliken distance (RMD); see [21,22,23,24] and the references therein. The exploration shows the commutativity of the permutations of the deprotonations. However, it does not necessarily cause the same energy after the optimization process, and the energy minima are not inherited.

Furthermore, shape theory has demonstrated similarities or differences between molecules resulting from deprotonation processes. In this way, in previous work [25], we were able to show that the lowest proton affinity between a set of molecules corresponds to the one that is the closest geometrically to the parent molecule. We use this formalism to establish comparisons between the geometries of the molecules and, with them, determine parallelisms with the deprotonation routes based on the pK_a_ values.

We consider the main reaction mechanisms of SET (single electron transfer) and SPLET (sequential proton-loss electron transfer) analyzed in an aqueous medium. From a thermodynamic point of view, it demonstrates which of them is the most favorable mechanism for compound Bd and molecular descriptors calculated concerning energy–orbital (Eo) methods: frontier orbitals, electronegativity (χ), hardness (η), electrophilicity (ω), and softness (S) [13].

Returning to anti-radical activity, we consider analyzing the interactions between Bd as the ligand and the enzyme aldose reductase (AR) as the receptor through docking and molecular dynamics. Aldose reductase (AR) is a cytosolic oxidoreductase that uses nicotinamide adenine dinucleotide phosphate (NADPH) as a cofactor. Moreover, it catalyzes the reduction of glucose to sorbitol, the first step in the polyol pathway of glucose metabolism. The next step in the pathway is catalyzed via sorbitol dehydrogenase (SDH), in which sorbitol is oxidized to fructose using nicotinamide adenine dinucleotide (NAD+) as a cofactor. This process impacts the NADPH/NADP+ ratio. NADPH is essential for regenerating the reduced form of the intracellular antioxidant glutathione [26]. Factors such as sorbitol accumulation and oxidative stress can complicate diabetes situations, and the inhibition of aldose reductase can prevent these secondary complications [27].

The aldose reductase reaction, particularly the production of sorbitol, is vital for the function of various organs in the body, such as the liver, lens, retina, and kidney [28]. Furthermore, the literature has reported studies where several antioxidant compounds inhibit this enzyme [29,30,31]. Finally, we employ our method based on shape theory formalism to find active sites on the enzyme.

Regarding pK_a_, this work includes, among others, the following items: 1. Describe the direct method in terms of a linear model that now better explains the pK_a_ in terms of the energy. 2. With the new model, we have been able to elucidate the deprotonation clusters using a combinatorial result. 3. We have made progress in predicting less than half of the calculations of all the routes in each round, an aspect that shows the importance of the newfound model. 4. Regarding the discrimination method with Riemannian geometry, we have defined a new distance that involves the Mulliken charges and the atomic masses. With the new measure, the geometric optimization is related to the deprotonation chemistry, and at the same time the pK_a_ is studied. The previous four elements have been written in a new model with their respective proofs.

This paper is organized as follows: In Section 2, Results and Discussion, we develop the studies: DFT analysis; prediction model for clusterization and computation of pK_a_, with the corresponding proof based on a chemical published argument; deprotonation route via pK_a_ and Riemann–Mulliken distance; exact linear model for pK_a_ in terms of energy; molecular docking; and molecular dynamics. In the methods section, we have explained the models and theories used in the calculations.

## 2. Results and Discussion

### 2.1. Analysis of the DFT Studies

Figure 1 shows that betanidin has three functional carboxyl groups (-COOH). Two are attached to the respective heterocycle through an sp3 atom (C2 and C15). Moreover, the third carboxylic group is attached to the C17 atom. Two hydroxyl groups (-OH) are connected to the catechol group at the C5 and C6 positions. Another intramolecular hydrogen bond is present at the dihydropyridine ring nitrogen. One end of the molecule bears a catechol group. The other cyclic amino group is joined via a π-conjugated system, generating an intramolecular electron transfer process [8,14]. The rings differ in the degree of delocalization. For example, the pyrrole ring is self-contained, but delocalization occurs in the carboxyl groups. This situation favors the movement of electric charges within the same molecule and antiradical activity [10].

The length of the bonds in Å between the oxygen atoms and their respective hydrogen susceptible to detaching from the molecule is the same for the case of carbonyl groups and very similar for the case of hydroxyl groups, for nitrogen, it is slightly more remarkable; the probability that these hydrogens will break off is quite close because the conditions are similar for each case. Atomic charges in the betanidin were calculated by employing the electronegativity–equalization method [32,33] and reported as a fraction of the electron’s electric charge measured in coulombs (Figure 2).

Additionally, the above-mentioned hydrogens also present similar partial charges and oxygens with which they bond. The electrostatic potential map analyzed the intramolecular interaction; Figure 3 illustrates the molecule’s charge distribution and reactive sites.

Subtle traces of shades of red in the vicinity of the oxygen atoms located at the extreme right of betanidin indicate a slight excess in electron density compared to the rest of the molecule, and the blue region at the other extreme indicates an excess of positive charge. The presence of conjugated systems with delocalization of the π orbitals and the subsequent probability of charge transfer can be increased by incorporating donor and acceptor substituents [14,34].

Other intermediate regions of the molecule and around the hydrogens of the carboxyl groups also show electron deficiency. Throughout the molecule, the electron densities are on average low or intermediate due to the presence of carbon atoms that have less electronegativity. This indicates that the electrostatic potential is lower (green); from right to left, along the delocalization, the electronic deficiency increases. The hydrogens bonded to the oxygens of the carboxyl groups indicate a possible site for a nucleophilic attack, most likely for the group located at the C15 and C17 positions. On the contrary, an electrophilic attack could occur in the hydroxyl groups, with a greater probability for the group found in C5. We notice that an electrophilic attack is less likely to occur than a nucleophilic attack.

#### 2.1.1. Reactivity Descriptors

We calculated and recorded (see Table 1) the basic properties of the electronic structure. We show the descriptors: ionization potential (vIP), vertical electron affinity (vEA), HOMO–LUMO gap (HLG), electronegativity (χ), hardness (η), electrophilicity (ω), and softness (S).

HOMO is the highest occupied molecular orbital. vIP is the negative of the (HOMO) energy, LUMO is the lowest empty molecular orbital, and vEA is the negative of the (LUMO) energy; the energy difference between the HOMO and LUMO is called the HOMO–LUMO gap. Bd presented a value of 2.41 eV, which indicates that charge transfer occurs within the molecule via the spacer system [34]. Therefore, the LUMO energy (−3.54 eV), HOMO energy (−5.95), and the gap value below 3 eV, Bd, can become an electron acceptor; see Figure 3.

Electronegativity indicates that the electron density of the system can vary. The calculated electronegativity value indicates that Bd can attract a charge to it. So, in an electron transfer opportunity, the Bd will accept and neutralize electrons from the radicals. For example, Bd has a value of 4.75 eV and indicates that the molecule tends to attract electrons. The hardness is related to the separation between the HOMO and LUMO. The smaller the energy gap, the smaller the resistance to change in electronic distribution, and vice versa. By presenting a hardness of 1.20 eV, the system under study implies a tendency to receive electrons. Softness measures the degree of chemical reactivity of the compound, which has a value of 0.83 eV and is the reciprocal of hardness. The high value of the electrophilicity descriptor (9.36 eV) confirms the deficiency of electrons. Thus, Bd is characterized by its high polarizability and by being an electron-accepting species.

This compound is not rich in electron density, indicating that electron-related mechanisms should not be responsible for antiradical activity. However, we evaluate the SET (single electron transfer) mechanism via the IP (ionization potential) descriptor. Furthermore, we compare it to the SPLET (sequential proton-loss electron transfer) mechanism associated with the PA (proton affinity) and ETE (electron transfer enthalpy) descriptors to verify which mechanism is more likely (Table 2).

When comparing the values of the descriptors, we observe that the values of PA and ETE are lower than IP. The thermodynamically most likely reaction pathway is the sequential transfer of electrons with loss of protons (SPLET). Furthermore, the lowest PA value is obtained for hydrogen from the C17-COOH group, followed by the C15-COOH and C2-COOH groups. For phenolic antioxidants, experimental and theoretical studies have also confirmed the importance of the SPLET mechanism in polar solvents [35]. Results show that the C17-COOH group deprotonates, followed by C2-COOH and C15-COOH, according to some reports [36,37], and followed by C15-COOH and C2-COOH, according to another author [20].

#### 2.1.2. Deprotonation Route and pK_a_-Values

Betalains’ free radical scavenging activity is pH-dependent [10,12,38] and, in aqueous solution, they have also been reported to be stable with a pH range of 3.5 to 7.0 [39]. In the literature, alterations in charge of betanin and isobetanin have been suggested, generating different forms of the compounds according to the pH of the medium, as follows: at a pH below 2, it presents a cationic form equal to 2, and a zwitterion form; between 2 and 3.5, it forms a mono-anion with deprotonated C2-COOH and C15-COOH groups; between 3.5 and 7 a dianion with deprotonated C2-COOH, C15-COOH, and C17-COOH groups; and at a pH between 7.0 and 9.5, a trianion appears with all carboxyl groups deprotonated and a phenolic C6-OH group [40,41]. Based on the solvent’s pH, betalains are found in several different states, and it is relevant to estimate the most likely state or form. Experimental pK_a_ values are known for some betalains. We mention betanin (Bn), composed of Bd and β-glucosidic linked to glucose at C5. We consider referenced-experimental pK_a_ values in the carboxyl groups. C2-COOH ranges from 1.1 to 1.5; C15-COOH and C17-COOH from 3.2 to 3.6; and for the phenolic OH group, from 8.5 to 8.9 [41,42].

The computational protocol DFT B3LYP/631+G(d,p) has been used widely since it allows for an analysis of the systems under study quite well, and the most stable conformers have been obtained in comparison with other protocols [10,20]. We calculated the pK_a_ values using the direct approach method described in the methodology and the deprotonation route for the carboxyl groups from these values. For a complete synchronization with the newly implemented linear model proved in Section 2.2, we have used our own optimization for the Bd parent, instead of the reference parent given in [20]. The new Riemann–Mulliken distance in Section 2.3 has enriched the conclusions of a purely geometric description using the Riemann distance. Thus, the subsequent analysis involves a new chemical measure with atomic mass and the Mulliken charge.

#### 2.1.3. Direct Approach

We consider Bn a reference and assume that Bd appears in the form of a zwitterion, mono-anion, and dianion. We calculated pK_a_ values using the direct approach method described in the methodology. Figure 4 shows the different states of mono-deprotonated Bd ordered from lowest to highest according to pK_a_ values.

According to the pK_a_ value, the first hydrogen cation (H^+^) starts from the C17-COOH group and is consistent with data reported by other researchers [20,36,37]. The theoretical data obtained are close to the experimental data reported in terms of experimental values taken as reference. However, there are differences when it comes to the group that deprotonates. To analyze the result, we considered the electrostatic potential surface of the molecule in its zwitterionic form according to the lowest pK_a_ (Figure 5).

We observe that the electric charge is delocalized in the double and conjugated bonds of the molecule. When the proton in any of the carboxyl groups is eliminated, the charge density is delocalized towards the site of the missing proton.. To a lesser extent, the hydrogen of the C17-COOH group causes the charge density to be affected less after removing the corresponding proton.

### 2.2. Prediction Model for Clusterization and Computation of pK_a_

Consider the first round of deprotonation for the Bd. According to [12,20,42] (see also our simulations), pK_a_ can be split in two groups: the inferior cluster of pK_a_ involves the carboxyl groups C15-COOH and C17-COOH (placed in the dihydropyridine ring nitrogen) and the carboxyl group C2-COOH (placed in the pyrrole ring). The superior cluster of pK_a_ considers the hydroxyl groups C5-OH and C6-OH (placed in the catechol ring), and the group HN16. The carboxyl groups are more acidic [12,20,42], and this can be one chemical justification for this clusterization.

Denote the inferior and superior clusters of the ih round as $I_{i,j},\;i=1,\;2,\;3;\;j=1,2,\ldots ,n_{i}/2$ and $S_{i,j},\;i=1,2,3;\;j=1,\;2,\;\ldots ,n_{i}/2$, respectively; where the subindex $j=1,2,\ldots ,n_{i}/2$, denotes the jth deprotonation route of round i, and $n_{1}=6,\;n_{2}=30,\;n_{3}=120$.

Based on the chemical results given in [12,20,42]:(1)$$pK_{a,1}(I_{1,j})<pK_{a,1}(S_{1,l})$$
for all j,l=1,2,3. 

Namely, the *pK_a_* of any deprotonation route of the inferior cluster in the first round is strictly less than any deprotonation route of the superior cluster in the same round. Therefore, the clusters of the first round are given by:(2)$$I_{1,.}=\left\{I_{1,j}:I_{1,1}=C2,I_{1,2}=C15,I_{1,3}=C17\right\}\;$$
(3)$$S_{1,.}=\left\{S_{1,j}:S_{1,1}=C5,S_{1,2}=C6,S_{1,3}=N16\right\}\;$$

Under the same regularity and chemical conditions of the of the direct approach, the predicted clusterizations for the second round are listed by the left multiplication of $I_{1,.}$ and $S_{1,.}$ with all the non-repeated elements of the those sets, namely:(4)$$I_{2,.}=\left\{I_{1,1}\;I_{1,.},I_{1,2}\;I_{1,.},\;I_{1,3}\;I_{1,.},\;S_{1,1}\;I_{1,.},S_{1,2}\;I_{1,.},\;S_{1,3}\;I_{1,.}\right\}$$
(5)$$S_{2,.}=\left\{I_{1,1}\;S_{1,.},I_{1,2}\;S_{1,.},\;I_{1,3}\;S_{1,.},\;S_{1,1}\;S_{1,.},S_{1,2}\;S_{1,.},\;S_{1,3}\;S_{1,.}\right\}$$

The same applies for the inferior cluster of the third route of deprotonation, where each element of $I_{1,.}$ and $S_{1,.}$ multiplies $\;I_{2,.}$ by the left element (without repeated elements). The superior cluster is also given by the left multiplication with $\;S_{2}$:(6)$$I_{3,.}=\left\{I_{1,1}\;I_{2,.},I_{1,2}\;I_{2,.},\;I_{1,3}\;I_{2,.},\;S_{1,1}\;I_{2,.},S_{1,2}\;I_{2,.},\;S_{1,3}\;I_{2,.}\right\}\;$$
(7)$$S_{3,.}=\left\{I_{1,1}\;S_{2,.},I_{1,2}\;S_{2,.},\;I_{1,3}\;S_{2,.},\;S_{1,1}\;S_{2,.},S_{1,2}\;S_{2,.},\;S_{1,3}\;S_{2,.}\right\}\;$$

**Law for clusterization prediction:** *In general, the deprotonation route*$D_{1}D_{2}\ldots D_{i-1}I_{1,j},\;i=1,\;2,\ldots 6;j=1,\;2,\;3.\;$*is located at the inferior cluster; for arbitrary previous deprotonations, i.e., every route finishing in*$I_{1,j},\;i=1,\;2,\;\ldots 6$*is placed in the inferior cluster of pK_a_. In the opposite way, the deprotonation route*$D_{1}D_{2}\ldots D_{i-1}S_{1,j},\;i=1,\;2,\ldots 6;j=1,\;2,\;3.$*gets higher pK_a_, and it is located at the superior cluster.*

Moreover, assume that $E\left(I_{1,j}S_{1,l}\right)\approx E\left(S_{1,l}I_{1,j}\right),\;l,j=1,\;2,\;3$ (see the simulations); in this case, the prediction of the high *pK_a_* values for$\;S_{2,.}$ can be obtained directly from the energies of $I_{2,.}$ by using the linear model:(8)$$pK_{a,2}(I_{1,j}S_{1,l})=mr-mE\left(I_{1,j}\right)+mE\left(S_{1,l}I_{1,j}\right)$$

If the *pK_a_* values for $S_{2}$ are available, then we can predict the values for $I_{2}$ by using the opposite model:(9)$$pK_{a,2}(S_{1,l}I_{1,j})=mr-mE\left(S_{1,l}\right)+mE\left(I_{1,j}S_{1,l}\right)$$

In the same manner, we can predict the superior (inferior) clusters of *pK_a_* by using the available energies under certain permutations, which are supported by the simulations.

Table 3 shows the equations for prediction of all the deprotonation routes based on the parallel linear models with positive constant slope m. As before, each pair of successive Equations ((10)–(11),(12)–(13),(14)–(15),(16)–(17)) is based on certain energy assumption supported by the data of the complete exploration of the deprotonation routes.

**Proof.** The proof of this result just follows from the linear model for pKa in terms of energies and the inequality for the corresponding energies:
$$E(I_{1,j})<E(S_{1,l})$$
for all j,l=1, 2, 3. See the Appendix A. □

**Remark 1.** *In practical situations, we do not require the application of all the pairs of models; rather, we just need to compute the energy in the expert software Gaussian for the lowest pK_a_. We prefer the energy computation of the inferior clusters (more stable), which will provide the best deprotonation route. Explicitly, the method recommends the computation in Gaussian of the following energies by applying the definition*:
$$pK_{a,3}=mr-mE_{2}+mE_{3}$$

And the approximations for lowest energies verified in the second round:

$E_{3}=E\left(S_{1,l}I_{1,j}I_{1,k}\right),\;l=1,\;2,\;3;j,k=1,\;2,\;3,\;j\neq k.$ and $E_{2}=E\left(S_{1,l}I_{1,j}\right)$ provide 18 inferior deprotonations.

$E_{3}=E\left(I_{1,l}S_{1,j}S_{1,k}\right),\;l=1,\;2,\;3;j,k=1,\;2,\;3,\;j\neq k.$ and $E_{2}=E\left(I_{1,l}S_{1,j}\right)\approx E\left(S_{1,j}I_{1,l}\right)$ provide 18 inferior deprotonations.

$E_{3}=E\left(I_{1,j}S_{1,l}I_{1,k}\right),\;l=1,2,3;j,k=1,\;2,\;3,\;j\neq k$ and $E_{2}=E\left(I_{1,j}S_{1,l}\right)\approx E\left(S_{1,l}I_{1,j}\right)$ via Equation (10) provide 9 deprotonations.

Finally, if we include the 6 computations of $E\left(I_{1,l}I_{1,j}I_{1,k}\right),\;l,j,k=1,\;2,\;3;\;l\neq j\neq k.$, then only 51 of the 120 possible deprotonations can be performed if the conditions of the linear model are accomplished.

There are several discrepancies in the best route of deprotonation. Reference [20] suggests different best deprotonation routes that are reached in the carboxyl group, i.e., C17C2C15 (see Figure 6). However, according to [10,20] the pathway starting in H-N16 is as possible as the referenced carboxyl routes.

It is easy to trace the source of this important assertion about H-N16 and C2-COOH in a number of papers. With the addressed discrepancies related to C17-COOH, H-N16 and C2-COOH, in general, there is not a consensus about the best deprotonation route.

We highlight, for example, the work [41] that supports the N16-H pathway by experiments. Just as other areas propose other options without moving away from the chemical context, they open the possibility of discovering different routes that are not classically possible. The references are open to all the options of deprotonation routes from a theoretical and/or experimental point of view.

However, suppose we extend our observation to other possible routes generated by the permutations of the order of the deprotonations. In that case, we can notice two aspects, one related to the property of pK_a_ commutativity and the other with the inheritance of the local minimum of energy between the different extraction levels. Both properties will not fulfill all the deprotonations.

This paper also advances the prediction of non-complete explored rounds. In particular, the model proved in Section 2.2 provides the location of higher deprotonation routes by using the clusterization of the first round. As can be seen in the statement after Equation (7), if we optimize an arbitrary deprotonation for 4, 5, and 6 rounds, we can predict the corresponding cluster. Given that the pK_a_ is just a simple linear model of the energy, then any associated stable and reasonable value can be promoted to a valid pK_a_. We submitted the addressed statement to a test of an interesting route, starting with a high pK_a_. Figure 7 and Figure 8 show the complete procedure of deprotonations, moving between superior and inferior clusters. The evolution of the pK_a_ is also described, and the final clusterization is obtained as we have expected; see also Table 4. This simple law is useful for avoiding unnecessary computations; we just need to select a priori deprotonation routes ending with a carboxyl group, no matter the initial location of the first or middle deprotonations.

For simplicity in the notation, and the left operation in the third column, we have removed the sub-indexes; see the left multiplication operation after Equation (7).

### 2.3. Alternative Clusterization of the Deprotonation Route via Riemann–Mulliken Distance

The literature does not explore all possible routes of deprotonation. It is assumed that the local minima of the first deprotonation are successively inherited. The energy or pK_a_ is related to the locus of the molecules. A variable can measure similarity or dissimilarity by using the geometry, the atomic mass, and the Mulliken charges. In this case we defined the Riemann–Mulliken distance (RMD) to compare the parent and the deprotonated configurations. It provides a descriptor using the recent theory of Riemannian geometry of the shape.

This matter is wholly contextualized in the optimization problem because a small change in the coordinates, mass, and charges of the molecules, concerning a reference, implies a low RMD. From the chemical point of view, these few changes can be a slight variation in the pK_a_. Therefore, we expect a relationship between the pK_a_ and the RMD. However, we do not enter into the details of this function; we will just provide some figures. We will research this topic in a future study.

We have computed all the Energy-pK_a_, and RMD-pK_a_ for the 156 deprotonation routes (6, first round; 30 second round, and 120, third round); see Appendix A, and see Appendix A.

The variable pK_a_ strongly depends on the energy; therefore, it is necessary to find an independent variable to study this affinity. Furthermore, an easy-to-measure predictor related to geometry, atomic mass, and Mulliken charges is also required. Due to this situation and the previous analyses, the RMD satisfies the requirements because it can detect minor differences in the geometric arrangement of a molecule and changes in the mass and charge. These variations translate into changes in physicochemical properties. Moreover, if we explore the entire energy surface, we can provide a non-statistical but algebraic-geometric total population space, which allows general conclusions about the deprotonation process. We start with the first deprotonation, and Figure 9 shows the claimed relation.

However, we have obtained two clusters for pK_a_, which are explained by a chemical argument [12,20,42]. The RMD is a good splitter for worst and better affinity. We notice that this separation is an expected behavior in the deprotonation process. We can apply several methods for clusterization, but according to the above-expected relation between pK_a_ and RMD of the parent (See Appendix A) and optimized molecules, we must demand an additional constraint for the two clusters; they must be parallel models. This restriction, which we will not consider in the classical methods for clusterization, motivates us to consider a new technique based on shape theory.

To get to this end, we used algebraic statistics of the six points mixed with the Riemannian geometry information of the population. Given that we expect two parallel linear models, we find all the possible subsets of four points and consider a simple rectangle as the template. The low Riemannian distance (RD) between the template and the four polygons will separate the clusters. The Riemannian distance is not invariant under the permutation of the landmarks. Then an additional step must involve orientable quadrilaterals. Figure 8 shows the orientable quadrilateral template. We have 15 possible orientable quadrilaterals compared with the template via Riemannian distance. Note that we have used the RD for the clusterization from a geometric point of view based on a large rectangle. The result is the same as the pK_a_ clusterization. However, we also use the enriched RMD for a geometric-chemical descriptor of the pK_a_. In this case, the clusterization is reached by grouping the extremal RMD shown in Appendix A.

The automatic detection by the RD gives two clusters: 1 (low): corresponding to the groups C2-COO-, C17-COO-, and C15-COO-; and 2 (high): C5-COO-, C6-COO-, and N16- as expected.

According to Figure 9, C17-COO- is the best first deprotonation and also obtains the minimum RMD. However, we do not consider any preferable way of getting the next best deprotonation. To get to this end, we consider the second deprotonation. In this case, we have 30 possible routes. Figure 10 shows the relationship between pK_a_ and the RMD between the parent and the deprotonated molecules.

Again, note that the clusters can be easily listed: Each permutation finishing in C2, C15, and C17 belongs to the inferior cluster; the remaining permutations belong to the superior clusters and end with C5, C6, and N16.

We can obtain the same clusterization by using the RD applied to the rectangular template of Figure 9. We compare the 27,405 oriented quadrilaterals with the same template of the first round of deprotonation.

As in the first deprotonation, we found two clusters. Therefore, we apply the clusterization method via a perfect template and the RD of all possible quadrilaterals. See classification in Appendix A.

Note that the pK_a_ is a linear function of the RMD. It is geometrically distinct on each cluster of the second deprotonation.

Also observed is that the first and second deprotonation numbers have been uniformly distributed in the two clusters. There is no preferable route of minimum energy, and the results will not transferfrom the given first deprotonation.

The best second deprotonation route includes C2-COO- and C17-COO-; however, C6-O-, C17-COO-, and C15-COO-, C17-COO- are also very near in pK_a_ and RMD. The table shows that the commutativity does not hold. For example, C17-COO- and C6-O- reach one of the highest pK_a_; several other issues can be inferred from both clusters (see Equations (8) and (9)). Finally, we considered the third deprotonation. Figure 11 again shows two differentiated clusters.

The inferior cluster (orange) is strictly constituted by all the permutations of the low cluster in the second round (dihydropyridine ring nitrogen: C2, C15, C17) and the superior and inferior cluster of the first round. The complementary set constitutes the superior cluster (blue). Note that the third deprotonation follows a rule that can be used for optimization of the computations. The superior cluster does not require computations, because we know that the pK_a_ will be high. According to the prediction model, we need to compute 51 of the 60 deprotonation routes of the inferior clusters (see Section 2.2).

The RD of 8,214,570 orientable quadrilaterals with the template gives the two clusters recognized in Appendix A. Again, the permutations of each triple provide a uniform distribution around the two clusters. Non-expert behavior can be traced by knowing only performance in the previous two deprotonations. Commutativity is also problematic. The best third deprotonation route corresponds to C6-COO-, C15-COO-, and C17-COO-. The prediction model of Section 2.2 guarantees this route because the last deprotonation corresponds to a carboxyl group. The reader can verify that all the routes of the inferior cluster finish in C2, C15, or C17.

Note also that the pK_a_ is linearly related to the RMD. We will study this interesting relation in future work.

Furthermore, the pK_a_ commutativity of deprotonation is not maintained. For example, if we switch the deprotonation of mono-anion C6-O-, C15-COO- in the second round, the behavior of the mono-anion C15-COO-, C6-COO- is opposite since it ranks in the high cluster. The respective pK_a_ values are 3.30 and 10.44. Appendix A show the energy differences under permutations. Only some equalities appear, supporting the hypothesis of the prediction models given in Section 2.2. Of course, this is a simple consequence of the prediction model in Section 2.2.

### 2.4. Exact Linear Model for pK_a_ in Terms of Energy

One of the main results of this work corresponds to the elucidation of pK_a_ as a simple linear function of energy. All deprotonation paths of any round are represented by straight lines on the pK_a_ energy plane. The linear functions are parallel to each other (with constant slope m) and only differ in the intercept, which in turn depends on the energy of the previous round. The literature reports different deprotonation routes for Bd. In some cases, it is difficult to determine the correct pK_a_ ranges; however, with the new linear model in terms of energy, the description of the deprotonation process simply depends on reasonable and stable energy. The calculation of the 156 deprotonations for rounds 1, 2, and 3 shows us that all the energies obtained are plausible and therefore the pK_a_ are reasonable. Appendix A show the pK_a_ in terms of energy for all rounds 1, 2, and 3 deprotonations. The corresponding linear models are shown in Figure 12, Figure 13 and Figure 14. The mathematical support for the model has been described in Section 3.4.

Note that the pK_a_ is divided into the same two clusters defined by the RMD. The best deprotonation path is located at the lowest energy of all, which in turn corresponds to the line with the highest intercept. The cluster prediction rule applies again here. If you want to know the cluster for each deprotonation route, it is enough to observe the last deprotonation. If the pathway ends in a carboxyl group, then it belongs to the lower cluster. If the route ends in a hydroxyl group or N16, it is located in the upper cluster.

This exact linear model (Section 3.4, derived from Equations (21)–(24)) is the support for the pK_a_ predictions that will be presented in the next section.

#### pK_a_ Predictions for Deprotonation Rounds 1, 2 and 3

In Section 2.2 we described a law for the prediction of the upper and lower pK_a_ clusters for any round. Table 3 of Section 2.2 also allows for the prediction of pK_a_ values from certain approximations between the energies of some subclusters. The 156 optimizations carried out in Gaussian support such suppositions.

Appendix A shows the dual pathway pK_a_ predictions for the second round. Figure 15 shows the excellent prediction of the respective subclusters in the second round and how the points are on a perfect prediction straight line with slope 1.

Appendix A and Figure 16, Figure 17, Figure 18 and Figure 19 show the various subcluster predictions for the third deprotonation. The equations that govern the predictions are detailed in Table 3 of Section 2.2. The predictions of the third round consider the results of the second round; therefore, the error is enlarged given the assumption for the equality of the energies. It is necessary to include three predictions for each route based on the minimum, the mean, and the maximum of the corresponding energies.

### 2.5. Molecular Docking

We carried out in silico molecular docking studies to identify binding residues in the active site of the enzyme aldose reductase (AR) in a complex with Bd. Applying this process, the binding affinity of the first ligand pose was −9.6 kcal/mol. The results show that the studied compound interacts favorably with the target through various non-bonded interactions, such as hydrophobic and hydrogen-bonding interactions. Figure 20 represents the close-up view of the AR active site bound with Bd in the binding pocket.

We can observe that four donor/acceptor hydrogen-bonding interactions are formed between the receptor residues Cys80, Leu300, His110, and Trp111 and carboxyl groups C17-COOH, C15-COOH, and C3-COOH, respectively. In addition, electrostatic pi–cation interactions occur between the dihydropyridine ring nitrogen and Trp111. Other hydrophobic or van der Waals contacts are formed by the spacer C12 of Bd and amino acids Trp111, Cys298, and Leu300. Finally, the catechol group presents a π-Alkyl with Val47. The carboxylate bound to C2 interacts with charged nicotinamide through hydrogen. In Table 5, we record information about the observed interactions.

Interactions of the complex AR enzyme with other ligands have been reported, consistent with our results. Therefore, with the residue, His110 appears in lidorestat [43], sulindac [44], citronellel [45], dehydrobenzoxazinone [29], and betanidin. The residue Leu300 appears in lidorestat, sulindac, and betanidin. The residue Trp111 corresponds to lidorestat [43] and betanidin. The residue Cys298 appears in sulindac [44], dihydrobenzoxazinone [29], and betanidin.

Figure 21 represents some physicochemical properties of binding through surfaces covering the pocket area of the protein and the ligand. In Figure 21a, the faint blue represents the active site’s basicity. Amber surfaces are regions with hydrophobic binding properties; the molecule occupies the receptor’s active site in the partly polar binding pocket primarily composed of Val47 andLeu300 (Figure 21b). Hot pink and lime green shades represent hydrogen bond donor and acceptor regions, respectively (Figure 21c).

To better understand the binding details and point out the relevance of medium acidity in an aqueous solution, we considered the charge alterations of Bd on pH changes proposed in the literature [42]. It was suggested that at 3.5 < pH < 7.0, a dianion with deprotonated C17-COOH, C15-COOH, and C2-COOH groups appears. Additionally, all carboxyl groups are dissociated in an environment of implicit water at physiological pH, and stabilized conjugated structures favor the type of scavenging reactions [12]. The molecular docking schema to analyze the interactions between the deprotonated Bd and the active site of the receptor can be seen in Figure 22.

The molecular docking schema to analyze the interactions between the deprotonated Bd and the active site of the receptor can be seen in Figure 22. The deprotonated carboxylate attached to the pyrrole ring interacts with NADP+ and forms an H-bond with Trp111 and His110. The other two deprotonated carboxylates also form H-bonds with Cys80 and Leu300. Residues Ala299 and Thr113 interact via H-bond with the ligand. Trp111 residue is in contact with the dihydropyridine ring’s nitrogen and the carboxylic group’s oxygen via π-cation and π-anion interactions, respectively. Hydrophobic interactions occur between the C12 spacer and the Cys298, Trp111, and Leu300 residues. Another hydrophobic interaction occurs between the catechol group and Val47. Table 6 includes information about the observed interactions.

Further, the binding affinity of the first ligand pose was −9.3 kcal/mol. The receptor amino acids interacting with the ligand are similar to those analyzed for the protonated molecule. However, both structures insert themselves deep into the pocket through mainly hydrogen bonding, electrostatic, and van der Waals interactions. However, it is not a coincidence that all the amino acids participate in these interactions.

Finally, we show the interaction between the deprotonated molecule (C6C15C17-COOH) and the protein’s active site. Figure 23 shows these interactions.

The binding affinity of the ligand was −9.5 kcal/mol. The interactions presented are of the hydrogen bond type between the ligand and the residues Trp111 (2.42 Å), Leu300 (3.10 Å), Cys303 (3.40 Å), and Tyr48 (3.50 Å). In addition, the NADP+ molecule also interacts via H-bond with Bd (2.38 Å).

In all cases, the interactions between the receptor and the ligand are consistent with those previously reported. Therefore, we selected the conformation of the AR-lidorestat complex. As a reference, according to our results and those previously reported, the molecule interacts with residues Cys80 and Trp111 through hydrogen bonds and with Leu300, Trp20, and Cys298 through van de Waals contacts. Additionally, it interacts with amino acids Trp79, Phe122, and His110 via another type of contact [29,43]. The results obtained from the AR-Bd complex’s behavior agree with those referred to as the AR-lidorestat complex.

#### Search for Active Sites in Automatic Molecular Docking via Shape Theory

We use a new method to automatically search protein pockets to study the protein-ligand system’s molecular coupling. This method allows us to determine a reference system with Riemann geometric information to analyze the receptor–ligand complex. It is based on the formalism of shape theory and a Lennard–Jones (L–J) potential 6–12 and 6–10. We reported the details of this method in a previous article (under review), and it was implemented in R, a free software environment for statistical computing and graphics [46]. We applied the algorithm for the automatic search of cavities in the protein and the classification of the pose of the ligand in the protein pocket according to the potential value. The ligands used correspond to betanidin in dianion form with deprotonated C17C15C2-COOH (Ligand 1) and C6C15C17-COOH (Ligand 2).

Figure 24 and Figure 25 show the cavity in which Ligand 1 and Ligand 2 geometrically fit with the corresponding pocket of the same protein and energetically present favorable interactions. Both figures show results consistent with those obtained in conventional software to perform molecular docking. The ligand is surrounded by dots representing the amino acid atoms surrounding the ligand, indicating favorable interactions between the ligand and the receptor’s active site. Although the method is purely geometric, it allows us to visualize the residues that come into contact with Ligand 1: Histidine 306, Tryptophan 219, 20, and 111, Cysteine 298 and 303, and Leucine 300. Ligand 2: Cysteine 80 and 303, Leucine 300, Histidine 110 and 306, Tryptophan 111, Phenylalanine 115, and Valine 47.

The approach has allowed us to identify the cavities and establish the interactions of the pose of each ligand, which we present in Table 7 and Table 8.

Our method automatically predicted the AR pocket and the possible interactions between the active site amino acids and the ligands (Ligand 1 and Ligand 2). In general, the interactions are hydrophobic for both cases with amino acids Trp111 ([29,43]) and Cys303 [47]. In Ligand 1, the deprotonated C17 carboxylate oxygen interacts electrostatically with His 306. This same interaction occurs between Trp219 [44] and the deprotonated C2 carboxyl group oxygen. For Ligand 2, it occurs similarly with the C17 carboxylate. This type of interaction is also generated with Trp219 for Ligand 1 and His110 for Ligand 2. Hydrogen bonds are present in Ligand 2 with Cys80 and Leu300 [43,47,48]. Cys80 and Val 47 [44] interact hydrophobically with the same ligand. Ligand 1 forms a hydrophobic interaction with NADP^+^ [43] and Trp20 [44].

Ligand 2, as mentioned before, represents a deprotonation pathway predicted by the geometric approach and located on the receptor and exhibits mainly π-alkyl type interactions at the catalytic site. The carboxylate groups are firmly attached to the active site by hydrogen bonding and electrostatic interaction [48]. Ligand 1 presents electrostatic interactions but not hydrogen bonds. Not much variation in protein conformation would be expected when interacting with Ligand 1 and Ligand 2, but moderate changes were presented regardless of the computational method used, by applying either conventional molecular docking or the one proposed by us. This situation may be because this protein has presented different binding site conformations for different inhibitors [43,44,49,50,51]. To investigate the flexibility of the aldose reductase binding site predicted by our method, we perform molecular dynamics (MD) simulations and report them below.

### 2.6. Molecular Dynamics

In this section, we present the molecular dynamics analysis applied to the receptor–ligand complex obtained by our method. In the Appendix A, we include this information for receptor–ligand models from conventional software for molecular docking calculations.

The molecular dynamics study’s objective is to simulate the local conformational changes in the complex that result from the binding between the protein pocket and the ligand. The procedure gives flexibility to the ligand and the protein residues to arrive at a conformation in a steady state. We chose for molecular dynamics the best conformation obtained from the molecular docking stage via our proposed method. We used the NAMD program, and the force field used was CHARMM36m, which can calculate parameters of bound or unbound molecules. Then we created an environment to put the molecule in to interact with the water. In this case, the protein is in a rectangular system with the correct amount of water molecules needed to solvate the box. We added KCl 0.15 M to the solution, which is necessary to eliminate electrostatic interactions between dissolved molecules.

At the end of the simulation, a file with the extension *.dcd was obtained, containing the residues’ trajectories during molecular dynamics. These trajectories are crucial to calculating the protein’s RMSD during the simulation, which is a widely used measure to quantify the structural deformation of the protein compared to the initial reference point. Figure 26a,b shows that the complexes formed with Ligand 1 (C17C15C2-COO-) and Ligand 2 (C6C15C17-COO-) during the molecular dynamics (DM) simulations have high stability during the simulation time. We can see that the structure does not vary significantly; there is an increase in the RMSD until reaching a point where the values fluctuate between 0.50 and 1.49 Å of RMSD (root mean square deviation), in the case of the protein associated with Ligand 1. For the other system, the RMSD value is between 0.44 and 1.77 Å. We can observe in Figure 26 that the deviations are within normality.

The average RMSD measure is similar for both structures: 1.14 and 1.32, respectively. Therefore, neither of the two complexes suffered a loss of structure during the simulation.

We calculated the deviation for Ligand 1 and Ligand 2 and presented a difference between the maximum and minimum value of RMSD below 5 Å, 0.95 and 1.01 for each molecule. Figure 27 shows the RMSD of Ligand 1 and Ligand 2. The average of this measure is similar for the two compounds, 0.94 and 1.37, respectively. For a system in equilibrium, the deviations have a maximum difference of 5 Å from RMSD, a situation fulfilled for these simulations, indicating that the compounds are in equilibrium.

Figure 28 depicts the potential interactions of the AR enzyme with Ligand 1 after molecular dynamics (last frame).

Figure 28 shows the amino acid residues of the active site interacting with the ligand. The green segmented lines show that the hydroxyl groups act as acceptors for forming hydrogen bonds, like the carboxyl groups of the dihydropyridine ring. We observe that Ligand 1 nests well on site after molecular dynamics and exhibits hydrophobic interactions with residues such as Trp20 (4.49 Å), Leu300 (5.24 Å), Trp11 (4.68 Å), and Cys298 (5.09 Å). Hydrogen bonds are with amino acids Leu300 (Å), Cys303 (Å), Cys80 (Å), and Ala299 (Å) [29,43].

Figure 29 shows the distances and interactions for the residues involved with Ligand 2. The hydrogen bond interactions correspond to residues Leu300 (2.29 Å), Cys303 (2.91 Å–2.5 Å), and Pro310 (2.69 Å). On the other hand, we can visualize a π-alkyl type interaction marked by a segmented pink line with the Trp111 (5.08 Å), Leu124 (Å), Leu300 (5.09 Å), Phe122 (5.22 Å), and Cys302 (4.67 Å) residues, as well as other hydrophobic interactions with Phe122 and Cys303 [29,43].

Once we apply molecular dynamics, the stability of each system obtained from our method [52] is verified, which is an alternative that automatically includes an exhaustive search of possible active sites.

Appendix A show that the complexes formed with Ligand 1 (C17C15C2-COO-) and Ligand 2 (C6C15C17-COO-) during the molecular dynamics (DM) simulations have high stability during the simulation time. Appendix A represent the potential interactions of the Ligand 1-AR and Ligand 2-AR complexes.

## 3. Materials and Methods

### 3.1. Computational Details

The capacity to scavenge free radicals through hydrogen atom transfer (H•) has been mainly approached through three mechanisms that are often discussed in the literature [13,53]. We consider the SET (single electron transfer) and the SPLET (sequential proton loss electron transfer) mechanisms.

The SET mechanism consists of one step defined by Equation (18). Here, an electron is transferred to a free radical, forming a radical cation.
(18)$$ArOH+R^{•}\to ArOH^{•+}+e^{-}$$

The SPLET mechanism consists of two steps (see Equations (19) and (20)). The first step refers to deprotonation, and the second is an electron transfer to a radical.
(19)$$ArOH\;\to ArO^{-}+H^{+}$$
(20)$$ArO^{-}\to ArO^{•}+e^{-}\;$$

The proton affinity (PA) and the electron transfer enthalpy (ETE) descriptors are shown to the SPLET mechanism and are obtained from PA = H_ArO_^−^ + H_H_^+^ − H_ArOH_ and ETE = H_ArO_^•^ + He^−^ − H_ArO_^−^, respectively. The ionization potential (IP) descriptor is identified to the SET mechanism and is calculated according to IP = H_ArOH_^•+^ + H_e_^−^ − H_ArOH_ [53].

Previous reports reveal the following: the enthalpy of the proton H(H^+^) is −259.00 kcal/mol; the enthalpy of the electron H(e^−^) is −55.61 kcal/mol; and the enthalpy of the hydrogen atom H(H^−^) is −314.65 kcal/mol. We consider these values and, additionally, all enthalpies were calculated at 298 K and 1.0 atm [53].

The computational study starts with a conformational analysis using the software ArgusLab [54] to identify the most stable structure. The optimization of the molecular geometry was performed by using the computational program Gaussian 09 package [55]. Examination of free radical-scavenging activity will be carried out with different protocols, and various reports have stated that the computational level DFT B3LYP/631+G (d, p) has allowed analysis of the system quite well at a low computational cost [10,20,53]. The geometries were optimized at the B3LYP level of theory and with the 6-31+G (d, p) basis set. The radical or ionic structures were optimized starting from the optimized geometries of the parent molecules by applying the available method UB3LYP/6-31+G (d, p), which is the SMD implicit solvation model [56]. The stability of the structures was confirmed by not obtaining imaginary frequency modes at the calculated vibrational frequencies. We determined the energies corresponding to the molecular orbitals of the HOMO/LUMO border, and from there, we obtained the values of the reactivity descriptors.

We find all possible permutations of the extractions of the hydrogens of interest in mono-deprotonation and subsequent deprotonations. It allows for the study of the different Riemann distances and their relationship with the deprotonation pathway and pK_a_ values. R is the package used in this step [46].

### 3.2. Deprotonation Routes and pK_a_ Values

Deprotonation routes have been studied, considering all possible combinations at each stage for each carboxyl and hydroxyl group within the molecule under study and their respective pK_a_ values. The pK_a_ values were calculated using different methods.

### 3.3. Direct Approach

Continuum solvent pK_a_ calculations utilize a thermodynamic cycle also known as the direct method, in which the deprotonation of the acid (HA^q^) in its conjugate base (A^q−1^) and the isolated proton (H^+^) are considered, and the acid charge HA^q^ is represented by q (See Equation (21)) [17,18].

This approach combines accurate gas-phase acidity with solvation-free energies from various solvent models. The free energy of the dissociation reaction of the acid in solution (∆G_aq_*) is calculated from the sum of the corresponding free energy of deprotonation in the gas phase of the acid (∆G_g_*) and the increase of free energies of solvation of the reagents and products involved (ΔΔG_solv_). The free energy in the gas phase of the dissociation reaction (∆G_g_*) is given by the expression observed in Equation (22).

The last term of numeral 1 corresponds to the energy variation associated with the change from the standard state from 1 atm to 1 M (∆G^1 atm →M^ = 1.89 kcal/mol). The value of the free energy in the gas phase of the proton at 298 K and 1 atm is −6.28 kcal/mol [18].

The increase in free energy of solvation between reactants and products can be calculated according to Equation (23).

This equation requires the value of the aqueous phase solvation free energy of the proton, ∆G_aq,solv_* (H^+^), which must be determined experimentally. The considered value of energy is −265.9 kcal/mol [57]. The superscripts in Equations (21)–(23), “*” and “^o^”, denote that the thermochemical quantities are computed with respect to a standard state of 1mol L-1 and 1 atm, respectively. From them, we arrive at the deprotonation energy of the acid AH^q^ in solution, and the corresponding *pK_a_* is calculated according to Equation (24), in which G_aq_*(A^q−1^) and G_aq_*(HA^q^) are the standard free energy of deprotonated and protonated species in the aqueous medium, respectively. All *pK_a_* calculations are given in the Appendix A.
(21)$$HA^{q}\left(aq\right)\overset{∆G_{aq}^{*}}{\to }A^{-}\left(aq\right)+A^{+}\left(aq\right)$$
(22)$$∆G_{gas}^{*}=G_{g}^{0}\left(H^{+}\right)+G_{g}^{0}\left(A^{q-1}\right)-G_{g}^{0}\left(HA^{q}\right)+RTLn\left(24.46\right)$$
(23)$$∆∆G_{solv}=∆G_{aq,solv}^{*}\left(H^{+}\right)+∆G_{solv}^{*}\left(A^{q-1}\right)-∆G_{solv}^{*}\left(AH^{q}\right)$$
(24)$$pK_{a}=\frac{∆G_{aq}^{*}}{RTLn\left(10\right)}=\frac{G_{g}^{0}\left(H^{+}\right)+∆G_{aq,solv}^{*}\left(H^{+}\right)+∆G^{1atm\to M}+G_{aq}^{*}\left(A^{q-1}\right)-G_{aq}^{*}\left(HA^{q}\right)}{RTLn\left(10\right)}$$

### 3.4. The Linear Model for pK_a_

In this work, Equation (24) is written as a linear model of the pK_a_ in terms of the deprotonation energy, as follows:$$pK_{a}=\frac{1}{RTln\left(10\right)}\left[G_{g}^{0}\left(H^{+}\right)+∆G_{aq,solv}^{*}\left(H^{+}\right)+∆G^{1\;atm\to M}\right]-\frac{1}{RTln\left(10\right)}\left[G_{aq}^{*}\left(HA^{q}\right)\right]+\frac{1}{RTln\left(10\right)}\left[G_{aq}^{*}\left(A^{q-1}\right)\right]$$

The slope of the model is:$$m=\frac{1}{RTln\left(10\right)}$$

There is a constant r for all the models:$$r=G_{g}^{0}\left(H^{+}\right)+∆G_{aq,solv}^{*}\left(H^{+}\right)+∆G^{1\;atm\to M}$$

Then the linear models is given by:$$pK_{a,i+1}=mr-mE_{i}+mE_{i+1},\;\;i=0,\;1,\;2.$$

Finally, the intercept-slope equation takes the form:$$pK_{a,i+1}=b_{i+1}+mE_{i+1},\;\;\;\;\;\;\;\;i=0,\;1,\;2.$$
$$b_{i+1}=mr-mE_{i},\;\;\;\;\;\;\;i=0,\;1,\;2$$
where $E_{i},\;i=0,\;1,\;2$ is the reference energy in kcal/mol for the deprotonation i+1 with energy $E_{i+1},\;i=0,\;1,\;2$. Given that the slope m is the same for all the rounds, then the intercept $b_{i+1},\;i=0,\;1,\;2$ defines parallel linear models for each $pK_{a,i+1}$ in terms of the independ variable $E_{i+1},\;i=0,\;1,\;2$. This implies that the deprotonation routes for a given round i+1, i=0, 1, 2 are easily shown in a graph of $pK_{a,i+1}\;\mathrm{vs}.\;E_{i+1},\;i=0,\;1,\;2$, where all the lines are parallel and can be ordered according to the intercept $b_{i+1},\;i=0,\;1,\;2$; therefore, the best deprotonation route is obtained in the lowest line.

Instead of discussion about reasonable $pK_{a,i+1}$, the above equations allow the treatment of all the deprotonation processes of every round i+1, i=0, 1, 2 as a simple problem of minimum energy selection. This quantity involves the physicochemical parameters associated with various deprotonation states [17,18,19] and the route to be carried out [20], a fact that makes a deprotonation reasonable if the corresponding deprotonated molecules are stable (negative reasonable energies).

In other words, this paper moves the study of the rounds of $pK_{a,i+1}\;$into the simple and acceptable world of stability and energy classification. As can be checked in the Appendix A, all the deprotonations in every round are reasonable because the corresponding energies are acceptable from the chemical point of view using the implemented theory.

### 3.5. Prediction Model for Clusterization and Computation of pK_a_

In the result section, we have provided a model that predicts the inferior and superior cluster of any deprotonation round based on the chemical clusterization for the first round via the direct approach of pK_a_.

### 3.6. The Riemann–Mulliken Distance

The geometric properties of pre-optimized (protonated) and optimized (deprotonated) molecules are not usually correlated. The literature does not reveal studies for clarifying the relationship between the resulting energy and the involved clusters before and after the optimized process of interest. In this case, we ask for a possible relationship from the geometry aspect based on a distance function between the molecules, including two important chemical elements: the atomic masses and the Mulliken charges. To achieve this end, we summarize the molecules in non-scaled matrices containing the x-y-z coordinates in the first three columns, followed by fourth and fifth columns with the atomic masses and the Mulliken charges. Each matrix represents the geometric and chemical information of the pre-optimized (protonated) and optimized (deprotonated) molecules. In this setting, the molecules are points in a quotient space (size-and-shape space), and distances among such points represent the similarity and dissimilarity of the parent and optimized molecules after deprotonation [21,25,52,58].

Given that the spatial coordinate matrices also involve Mulliken’s charges and atomic masses, then the Riemann–Mulliken distance (RMD) is appropriate for registering the discrepancies in the deprotonation process. We do not consider the RDM as a new chemical method for characterization of pK_a_; rather, we open the perspective to future research, moving the analysis of deprotonation to some interesting quotient spaces susceptible of useful invariances for the optimization process. Explicitly, we ask for a possible relationship in the RMD distribution of all possible extractions of a parent molecule with the corresponding distribution when the parent is protonated. The mathematics behind the RDM can be seen in the context of the well-known Riemannian distance in size and shape spaces provided by [21], among others. A number of applications in chemistry, physics, engineering, and biology have used similar concepts on Riemannian theory; see, for example, [21,22,23,24,25,52].

In this preliminary work about the emergent RMD, we have noted that some linear relationships with the pK_a_ can be explored in the future. Finally, we used the free software for statistical computing R [46] to implement our algorithms, which contain the new RMD and the linear models for prediction and clusterizations of the pK_a_.

### 3.7. Molecular Docking

For the development of this study, we selected the enzyme aldose reductase (AR). Its importance lies in the fact that the polyol pathway contributes to the production of oxygen reagents and can generate oxidative stress. This pathway is activated when the glucose concentration becomes too high in the cell; in this case, aldose reductase catalyzes the conversion of glucose into sorbitol, and sorbitol is oxidized to fructose. The accumulation of sorbitol increases the osmotic pressure, and this situation can participate in the progress of various complications associated with diabetes [59].

We explored the possible interactions between the enzyme aldose reductase (AR) and the compound under study through molecular modeling using the AutoDock Vina version program [58]. In molecular docking studies, we used the crystal structure of aldose reductase in complex with tolmetin, whose code is 1Z3N (Protein Data Bank: http://www.rcsb.org/pdb, accessed on 1 January 2022) [43]. Polar hydrogens were attached to the protein atoms, then partial Kollman charges of bonded atoms were assigned. We built and optimized the 3D structure of the Bd compound with Gaussian software [30]. To carry out the docking simulations, we defined a grid box with dimensions 18 Å × 18 Å × 18 Å and a center set at the point x = 15.0, y = −7.0, z = 11.0 to enclose the reported active site [47]. We selected the representation of the most favorable binding mode predicted by the AutoDock Vina calculations according to the coupled model with the best score (lowest coupling energy) and analyzed it with the VMD (Visual Molecular Dynamics) [60] and Discovery Studio programs [61]. With an algorithm that we implement in R [46] and that is based on shape theory, we carry out the automatic search for active sites and analyzed the docking. This proposed model is detailed in a previous work that has been submitted [52].

### 3.8. Molecular Dynamics

We carried out the molecular dynamic calculations with NMAD (Nanoscale Molecular Dynamics), a molecular dynamics package designed for high-performance simulations of movement of biomolecules over time [62], and CHARMM-GUI (Chemistry at Harvard Macromolecular Mechanics-GUI), a web-based graphical user interface for generating various input files and standardization of atomic coordinate and dynamic trajectory analysis and manipulation techniques [63]. We used one of the conformations with the highest score obtained from the molecular docking coupling for molecular dynamics calculations. The graphical user interface that we used has the CHARMM36m force field integrated, which we used with the TIP3P water model [64].

The simulation ran in 2.5 ns. The first phase of equilibration with NVT assembly aims to stabilize the system temperature; later, in the second phase with NPT assembly, the system pressure is stabilized via the periodic boundary conditions with the Particle–Mesh Ewald (PME) method to model electrostatic effects throughout the simulation. The simulations were run in a one-step heating process at a temperature of 303.15 K. All simulation steps’ time-step were set to 2 fs.

We used the VMD and Discovery Studio programs to analyze the protein–ligand complex, molecular dynamics (MD) simulation trajectory, and ligand–enzyme interactions. At the end of the simulation, we obtained a file with the extension *.dcd, which contains the trajectories of the residues during the molecular dynamics.

The corresponding video of the molecular dynamics can be seen at Ramirez Velásquez, Iliana; Bedoya-Calle, Álvaro H.; Velez, Ederley; Caro-Lopera, Francisco J. (2022), “Dissociation Mode of the O-H Bond in betanidin, pKa-clusterization Prediction and Molecular Interactions via Shape Theory and DFT Methods”, Mendeley Data, V5, www.doi.org/10.17632/hdrgfbzjcn.5, see Appendix A.

## 4. Conclusions

Based on a linear model of pK_a_ in terms of energy, this work has proposed a prediction law for the upper and lower clusters. The 156 optimizations of the first three rounds have ratified the assumptions of the model and have allowed for double prediction of the pK_a_ values.

We show that the local minimum of energy is not inherited by successive deprotonation processes. It is possible to find a local minimum in posterior deprotonations starting from a high pK_a_ in the first deprotonation. Commutativity is not also holding for these processes.

All the simulations in Bd based on the linear model of the pK_a_ are supported by a published chemical argument that allows for the prediction of the superior or inferior clusters. In terms of the groups of the Bd, the model established that if the last deprotonation includes a hydroxyl group or NH16, then the route is placed in the superior cluster of high pK_a_. Otherwise, if the last deprotonation involves a carboxyl group, then the route belongs to the inferior cluster of low pK_a_. The addressed law was verified in all the possible 6, 30, and 120 deprotonation routes of rounds 1, 2, and 3, respectively. It held also in some random deprotonation routes of the fourth and fifth round.

Our computational studies have allowed us to conclude that the thermodynamically most likely reaction pathway for betanin is sequential electron transfer with loss of protons (SPLET). In addition, the lowest PA value is obtained by hydrogen from the C17-COOH group, followed by the C15-COOH and C2-COOH groups.

One of the possible uses of predictions based on the linear model and the RMD descriptor enables in situ spectroscopy of biological substances in an indirect measurement of pH assisted by fluorescence spectroscopy.

The linear model described for pK_a_ and its predictions for Bd can be used as a biological marker or fingerprint of the substance under study.

The analysis of the behavior of protonated and deprotonated betanidin in the active site of the enzyme aldose reductase shows that it interacts with residues Trp111, His110, Cys298, Leu300, Val47, Trp20, Phe122, and Trp20 and involves van der Waals, H-bond, and electrostatic interactions. These results are consistent with previously published reports.

When applying our method in searching for cavities in the AR enzyme and classifying deprotonated Bd ligand poses, it was possible to predict the pocket in which the previously analyzed interactions occurred. In addition, it has been possible to report other cavities that have not been studied in the biological field.

Betanidin is a promising material in green synthesis design as it has an HLG of 2.44 eV for solar cells that corresponds to an acceptor semiconductor.

## Figures and Tables

**Figure 1 ijms-24-02923-f001:**
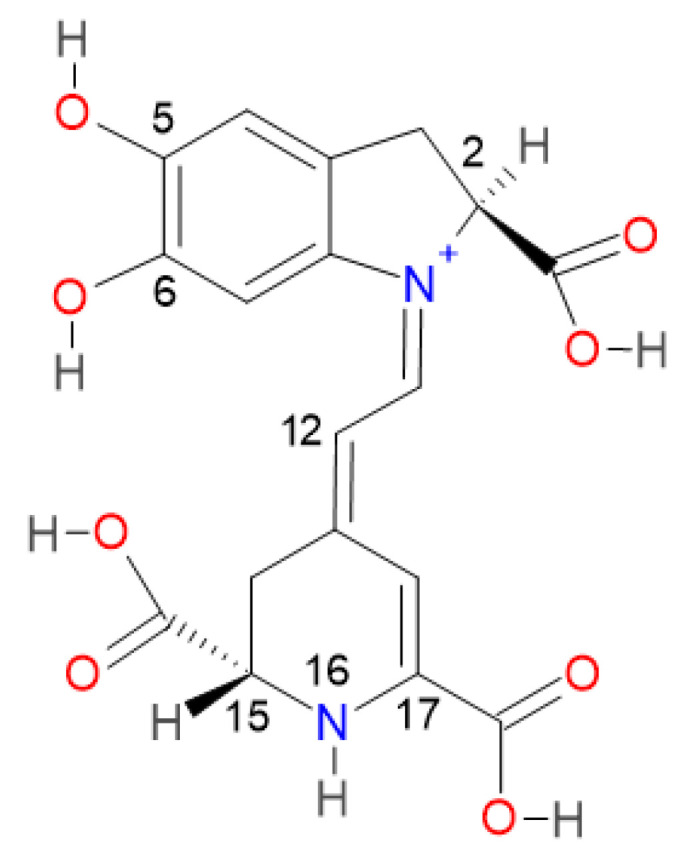
Chemical structures and atom numbering system of betanidin (Bd).

**Figure 2 ijms-24-02923-f002:**
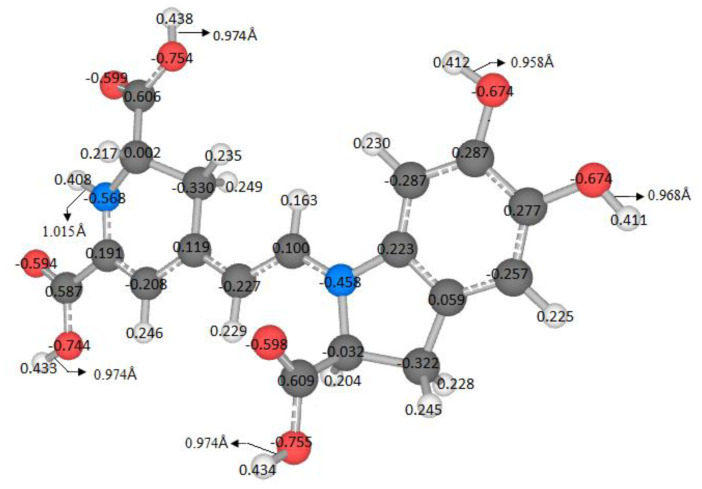
Partial charges in terms of the charge of the electron in C and bond lengths in Å.

**Figure 3 ijms-24-02923-f003:**
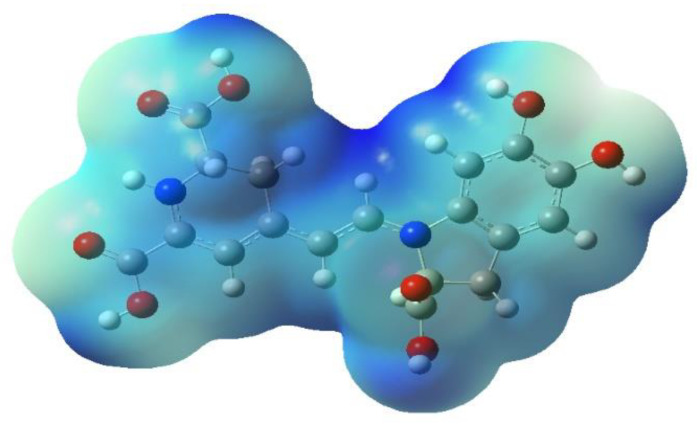
Map of electrostatic potential for Bd (isovalue = 0.02 density = 0.0004).

**Figure 4 ijms-24-02923-f004:**
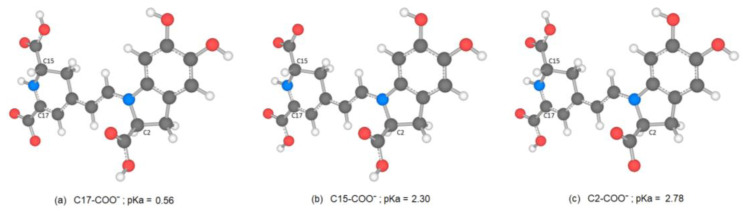
Optimized mono-deprotonated structures (**a**–**c**) in the three carboxylic positions and calculated corresponding pK_a_-values.

**Figure 5 ijms-24-02923-f005:**
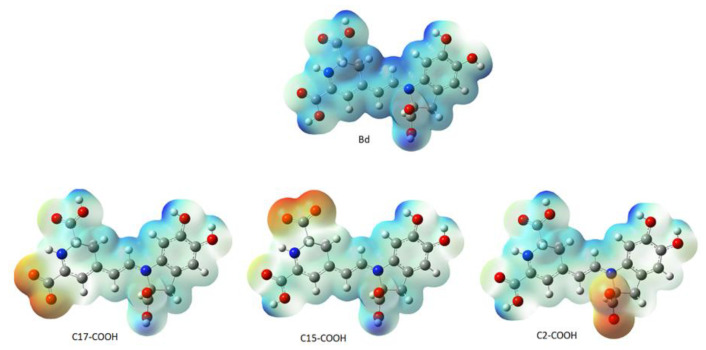
Electrostatic potential surfaces of Bd and zwitterionic forms.

**Figure 6 ijms-24-02923-f006:**
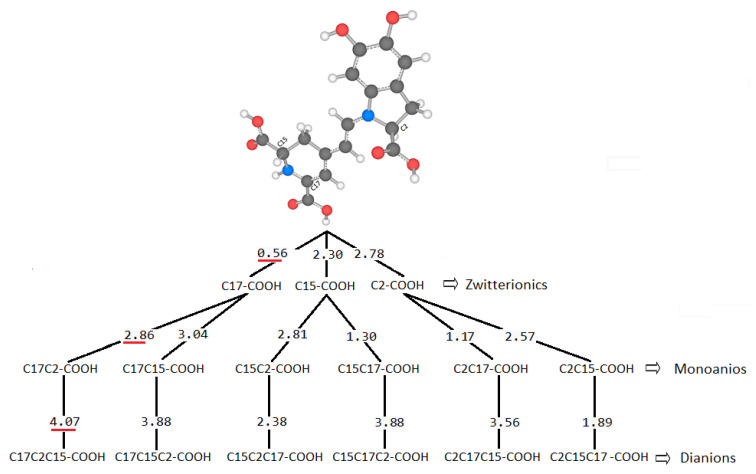
Calculated pK_a_-values and deprotonation route for carboxyl groups via direct approach.

**Figure 7 ijms-24-02923-f007:**
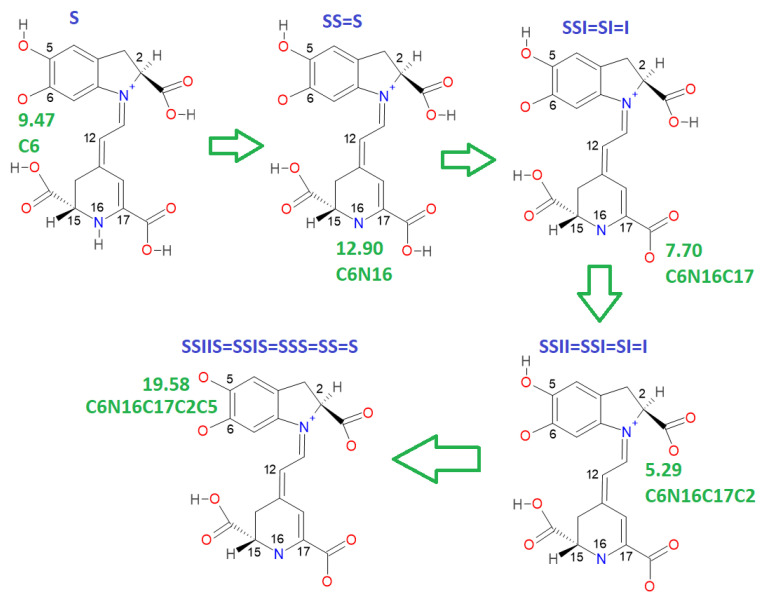
Cluster prediction location of a fifth deprotonation route starting with a very high pK_a_; the evolution in the rounds clearly jumps between superior and inferior clusters. According to the statement proved in Section 2.2, if the last deprotonation includes a hydroxyl group or NH16, then the route is placed in the superior cluster; otherwise, if the last deprotonation belongs to a carboxyl group, then the route is in the inferior cluster of low pK_a_.

**Figure 8 ijms-24-02923-f008:**
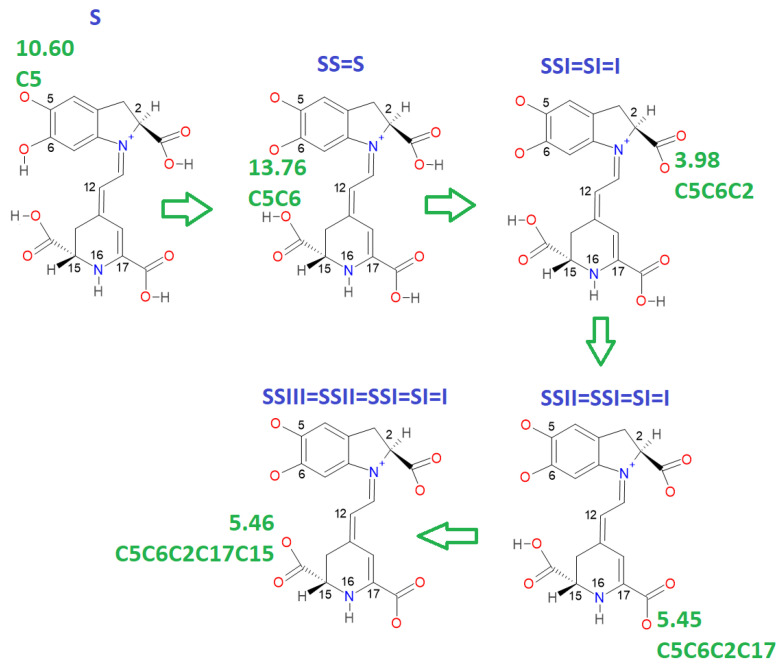
A different deprotonation route involving low pK_a_ in the last stage. The pathway starts on a high value in the first round. The proved statement of Section 2.2 guarantees the result; if the last deprotonation includes a carboxyl group, then the route belongs to the inferior cluster.

**Figure 9 ijms-24-02923-f009:**
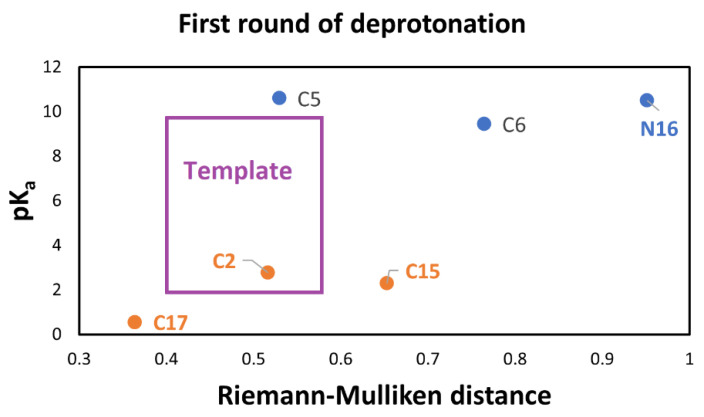
Relation between RMD and pK_a_ values of Bd, and template for clusterization via RMD. The low cluster involves the dihydropyridine ring nitrogen (C2, C15, C17), and the high cluster corresponds to the catechol ring (C5, C6, N16).

**Figure 10 ijms-24-02923-f010:**
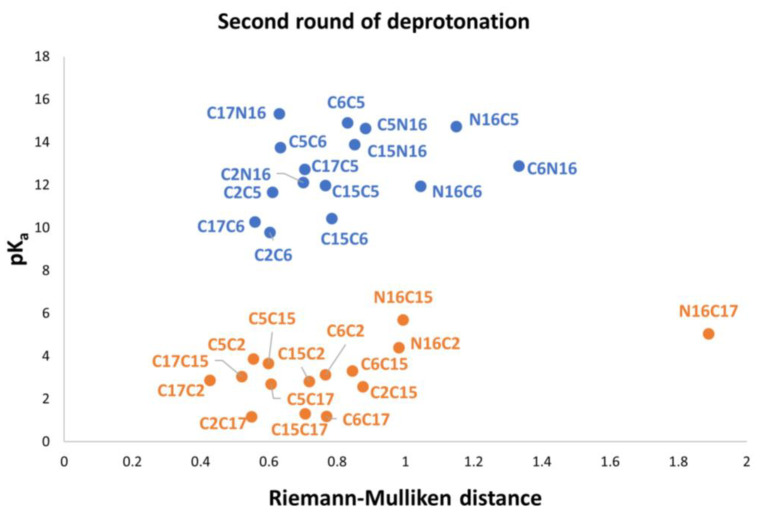
Relation between RMD and pK_a_ values for all possible second deprotonations in Bd. The inferior cluster (orange) is strictly constituted by all the permutations of the low cluster in the first round (dihydropyridine ring nitrogen: C2, C15, C17). This cluster also involves all possible combinations of the superior group of the first round, catechol ring: C5, C6, and N16; followed by all possible combinations of the low cluster in the first round. The complementary set constitutes the superior cluster (blue). Note that the second deprotonation follows a rule that can be used for optimization of the computations. The superior cluster does not require computations because we already know that the pK_a_ will be high. However, note that the inferior cluster must be computed completely because the addressed rule cannot provide the best route.

**Figure 11 ijms-24-02923-f011:**
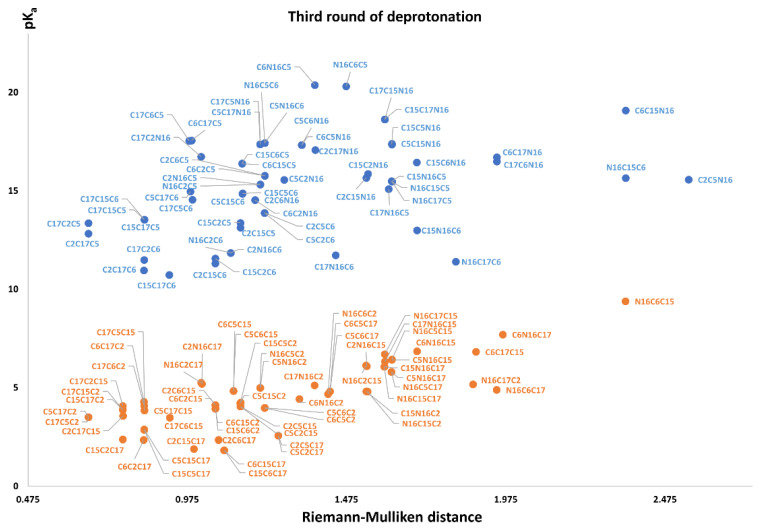
Relation between RMD and pK_a_ values for all possible second deprotonation in Bd.

**Figure 12 ijms-24-02923-f012:**
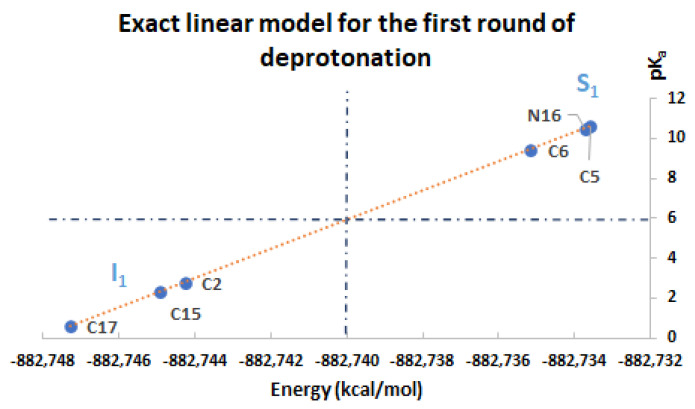
pK_a_ calculation used a direct approach for the first round of deprotonation. Two subclusters are formed: I_1_ with C2, C15, and C17; S_1_ with C5, C6, and N16.

**Figure 13 ijms-24-02923-f013:**
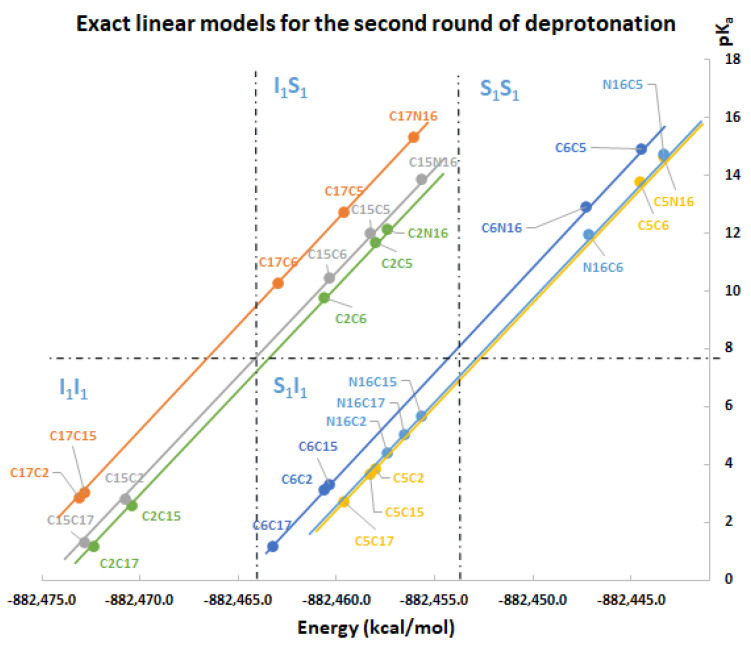
pK_a_ calculation used a direct approach for the second round of deprotonation. Four subclusters are formed: I_1_I_1_, S_1_I_1_, I_1_S_1_, and S_1_S_1_.

**Figure 14 ijms-24-02923-f014:**
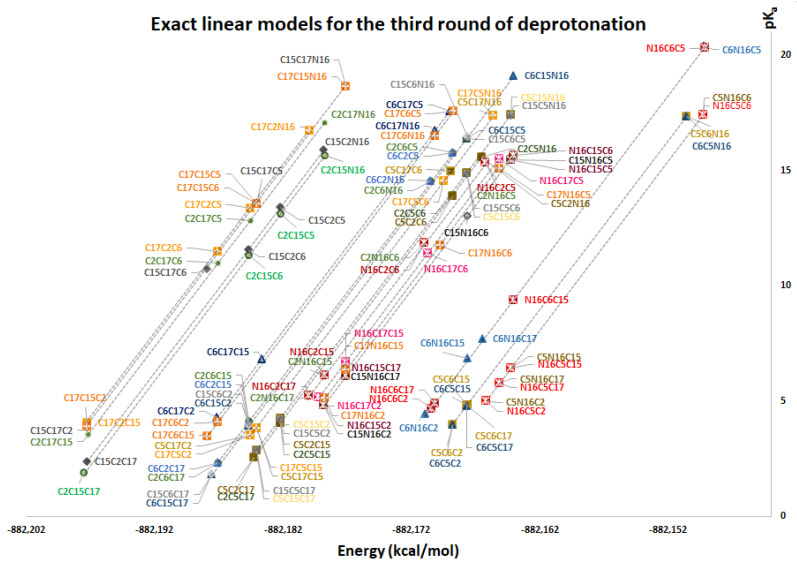
pK_a_ calculation used a direct approach for the third round of deprotonation.

**Figure 15 ijms-24-02923-f015:**
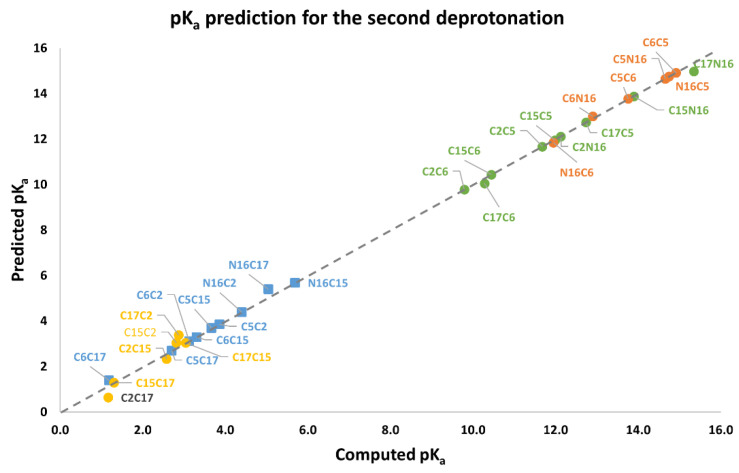
pK_a_ calculation using a direct approach for the first round of deprotonation. Two subclusters are formed: I_1_, with C2, C15, and C17; S_1_, with C5, C6, and N16.

**Figure 16 ijms-24-02923-f016:**
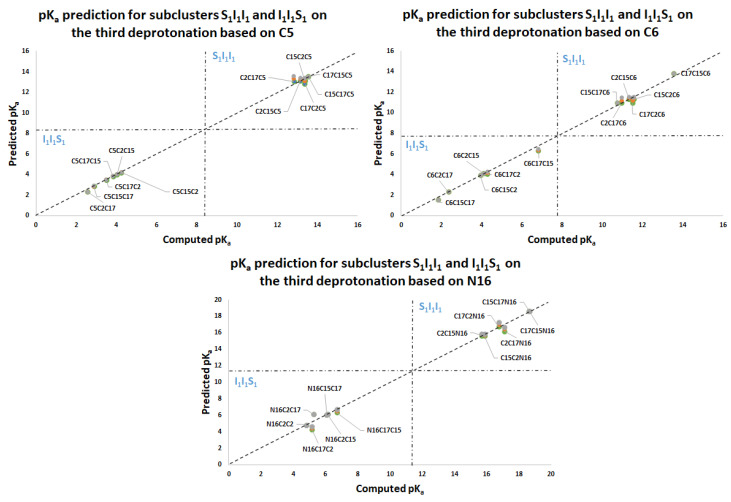
pK_a_ prediction for the third deprotonation on the subclusters S_1_I_1_I_1_ and I_1_I_1_S_1_ based on C5, C6, and N16. Predictions are marked as gray (max), orange (mean), and green (min).

**Figure 17 ijms-24-02923-f017:**
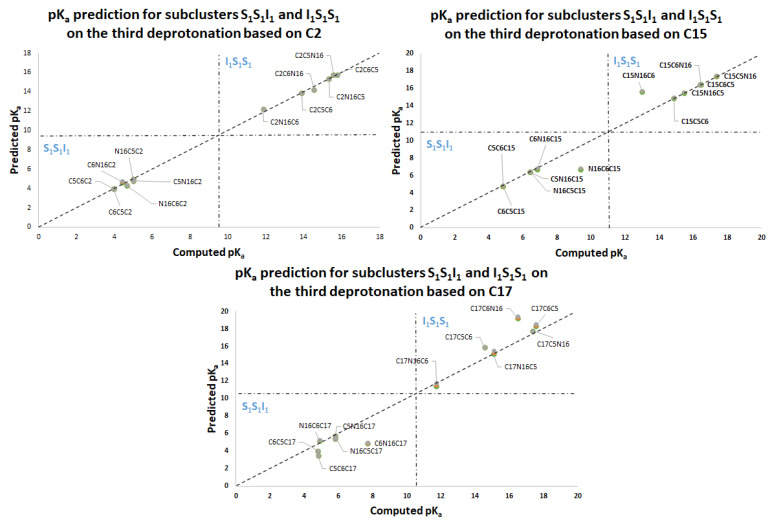
pK_a_ prediction for the third deprotonation on the subclusters S_1_S_1_I_1_ and I_1_S_1_ S_1_ based on C2, C15, and C17. Predictions are marked as gray (max), orange (mean), and green (min).

**Figure 18 ijms-24-02923-f018:**
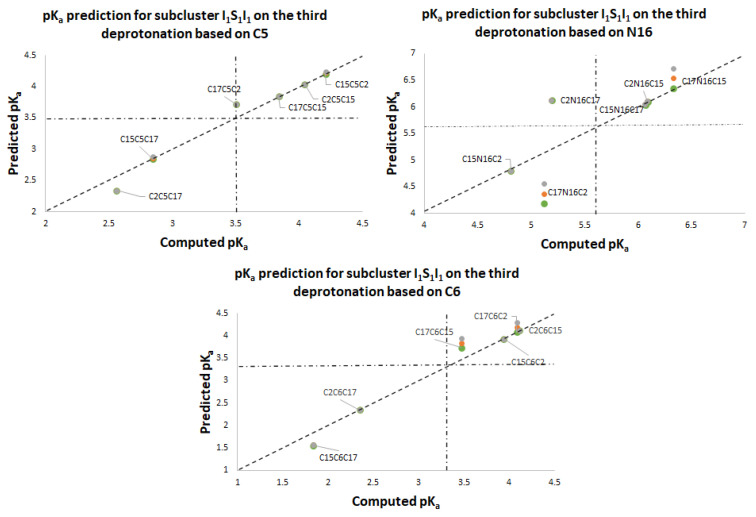
pK_a_ prediction for the third deprotonation on the subclusters I_1_S_1_I_1_ based on C5, N16, and C6. Predictions are marked as gray (max), orange (mean), and green (min).

**Figure 19 ijms-24-02923-f019:**
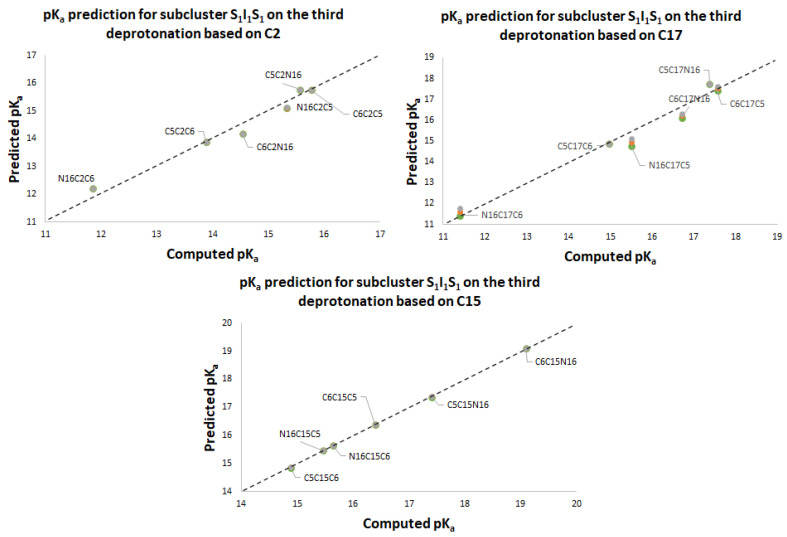
pK_a_ prediction for the third deprotonation. Subclusters S_1_I_1_S_1_ based on C2, C17, and C15. Predictions are marked as gray (max), orange (mean), and green (min).

**Figure 20 ijms-24-02923-f020:**
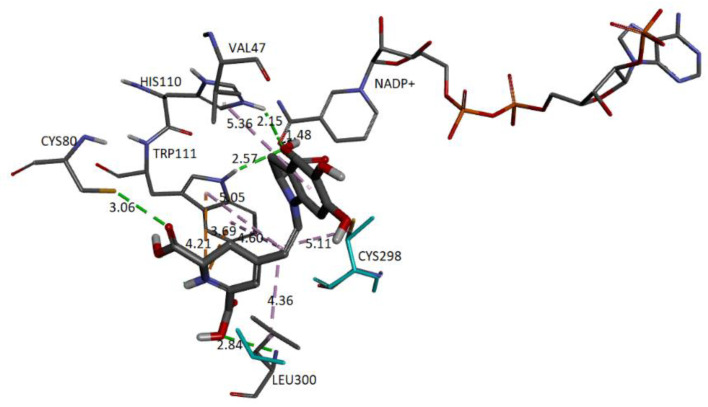
AR structure and molecular basis of potential active site interactions with Bd.

**Figure 21 ijms-24-02923-f021:**
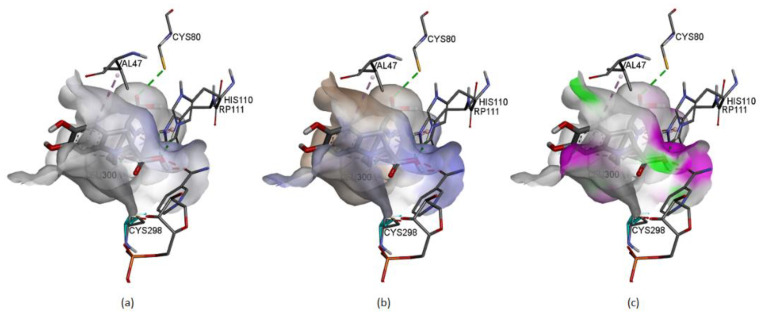
(**a**–**c**) represents the physicochemical properties of binding through surfaces.

**Figure 22 ijms-24-02923-f022:**
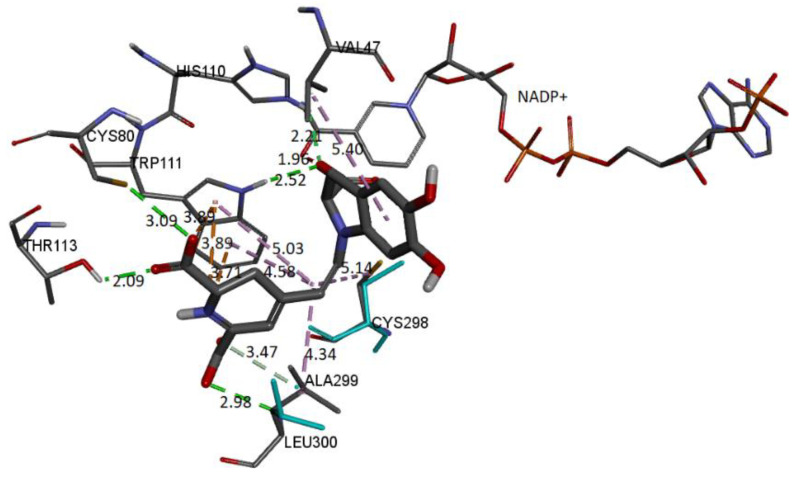
Molecular basis of potential active site interactions with deprotonated Bd (C17C15C2-COOH).

**Figure 23 ijms-24-02923-f023:**
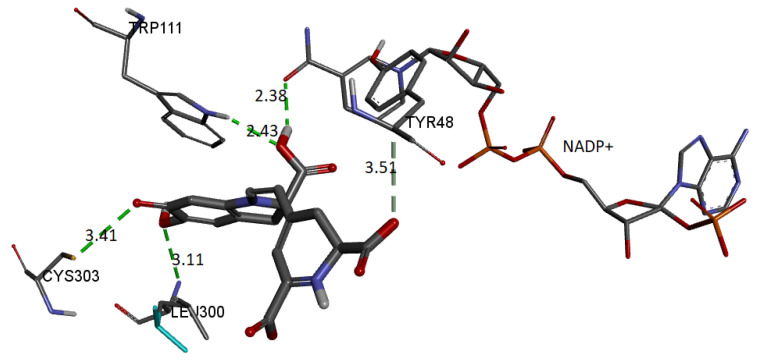
Molecular basis of potential active site interactions with deprotonated Bd (C6C15C17-COOH).

**Figure 24 ijms-24-02923-f024:**
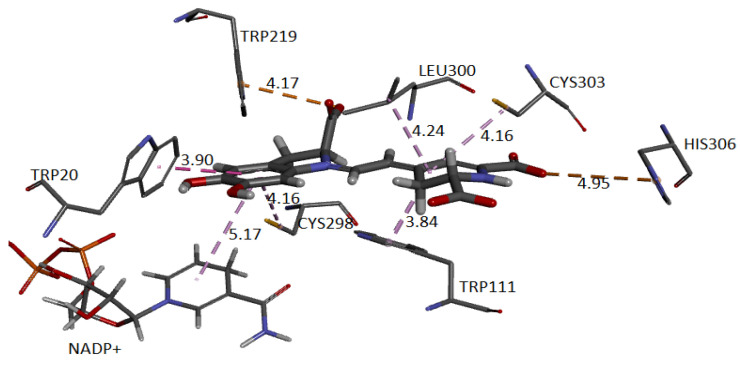
Molecular basis of potential active site interactions with deprotonated Bd (C17C15C2-COOH) via our method.

**Figure 25 ijms-24-02923-f025:**
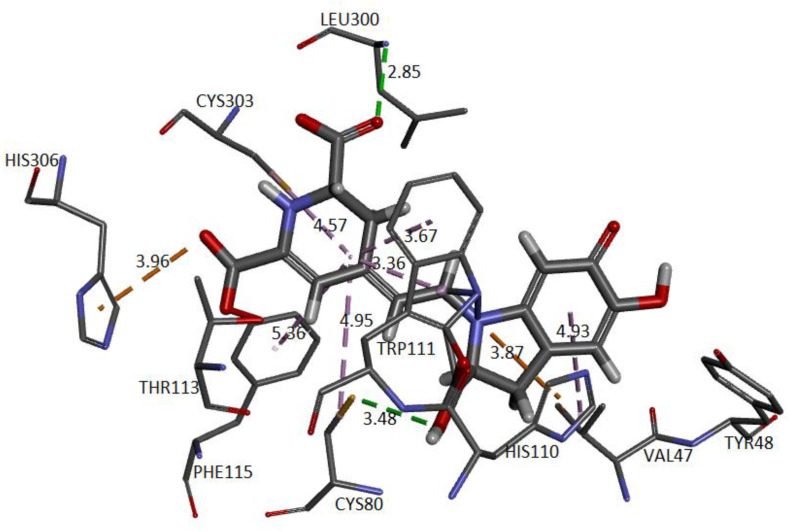
Molecular basis of potential active site interactions with deprotonated Bd (C6C15C17-COOH) via our method founded on shape theory.

**Figure 26 ijms-24-02923-f026:**
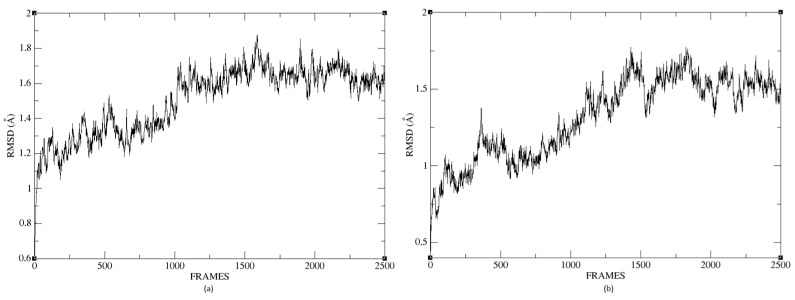
RMSD of the protein backbone during simulation of MD trajectories, (**a**) for the protein related to Ligand 1, and (**b**) for the protein related to Ligand 2.

**Figure 27 ijms-24-02923-f027:**
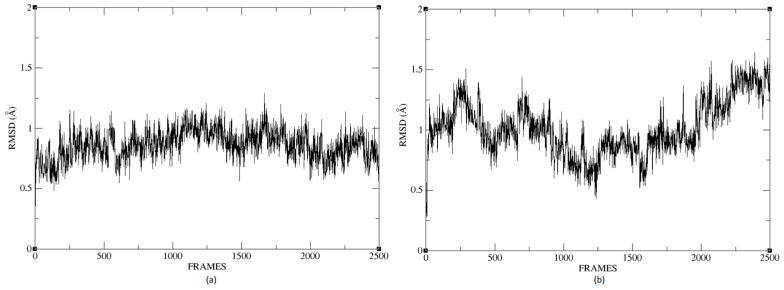
RMSD of the compounds during simulation of MD trajectories: (**a**) Ligand 1 and (**b**) Ligand 2.

**Figure 28 ijms-24-02923-f028:**
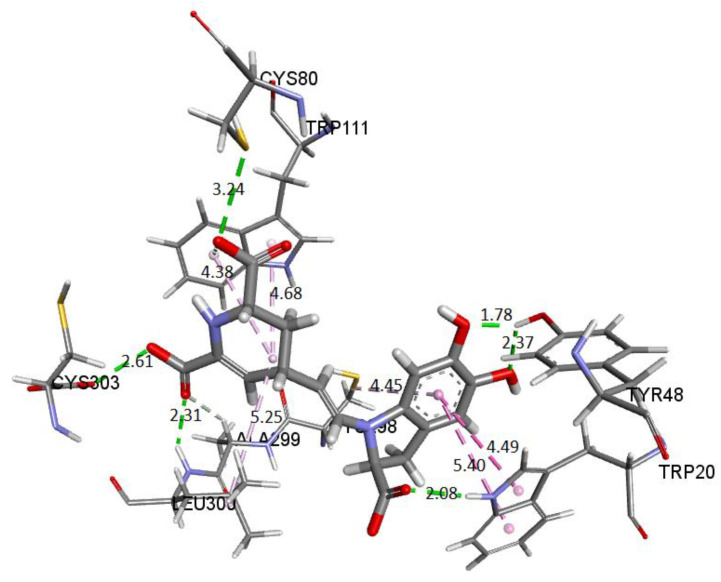
The DM study obtained the interactions of π and hydrogen bonds of the Ligand 1 compound with AR.

**Figure 29 ijms-24-02923-f029:**
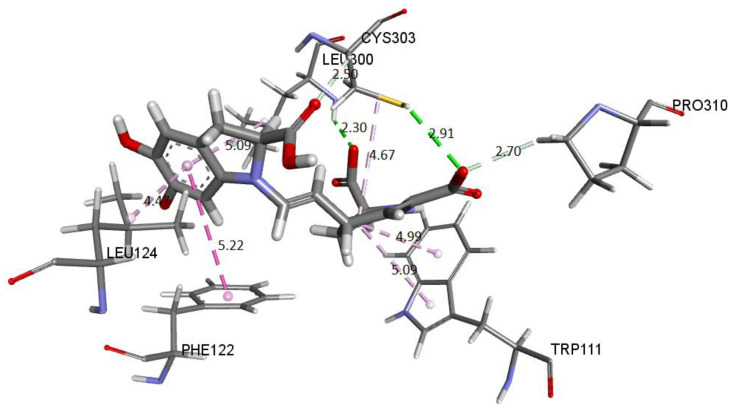
The DM study obtained the interactions of π and hydrogen bonds of the Ligand 2 compound with AR.

**Table 1 ijms-24-02923-t001:** Molecular descriptors of Bd.

Descriptors	Units (eV)
vIP	5.95
vEA	3.54
HLG	2.41
Electronegativity (χ)	4.75
Hardness (η)	1.20
Softness (S)	0.42
Electrophilicity (ω)	9.36

**Table 2 ijms-24-02923-t002:** Molecular descriptors of Bd: IP, PA, and ETE.

Bond	SET	SPLET	
	IP	PA	ETE
C2	75	14.9	70.4
C15		14.2	70.6
C17		11.6	68.6
N16		23.2	60.6
C6		22.9	50.4
C5		25.8	49.6

**Table 3 ijms-24-02923-t003:** Equations for prediction of all deprotonation routes based on the parallel linear models.

Verified Assumption	Dependent Variable	Intercept	Independent Variable	Cluster	Eq.
$E\left(I_{1,j}S_{1,l}I_{1,k}\right)\approx E\left(I_{1,k}S_{1,l}I_{1,j}\right),\;l=1,2,3;j,k=1,2,3,\;j\neq k.$	$pK_{a,3}(I_{1,j}S_{1,l}I_{1,k})$	$mr-mE\left(I_{1,j}S_{1,l}\right)$	$E\left(I_{1,k}S_{1,l}I_{1,j}\right)$	lower	(10)
	$pK_{a,3}(I_{1,k}S_{1,l}I_{1,j})$	$mr-mE\left(I_{1,k}S_{1,l}\right)$	$E\left(I_{1,j}S_{1,l}I_{1,k}\right)$	lower	(11)
$E\left(S_{1,j}I_{1,l}S_{1,k}\right)\approx E\left(S_{1,k}I_{1,l}S_{1,j}\right),\;l=1,2,3;j,k=1,2,3,\;j\neq k.$	$pK_{a,3}\left(S_{1,j}I_{1,l}S_{1,k}\right)$	$mr-mE\left(S_{1,j}I_{1,l}\right)$	$E\left(S_{1,k}I_{1,l}S_{1,j}\right)$	higher	(12)
	$pK_{a,3}\left(S_{1,k}I_{1,l}S_{1,j}\right)$	$mr-mE\left(S_{1,k}I_{1,l}\right)$	$E\left(S_{1,l}I_{1,l}S_{1,k}\right)$	higher	(13)
$E\left(S_{1,l}I_{1,j}I_{1,k}\right)\;\approx E\left(I_{1,j}I_{1,k}S_{1,l}\right),\;l=1,2,3;j,k=1,2,3,\;j\neq k.$	$pK_{a,3}\left(S_{1,l}I_{1,j}I_{1,k}\right)$	$mr-mE\left(S_{1,l}I_{1,j}\right)$	$E\left(I_{1,j}I_{1,k}S_{1,l}\right)$	lower	(14)
	$pK_{a,3}\left(I_{1,j}I_{1,k}S_{1,l}\right)$	$mr-mE\left(I_{1,j}I_{1,k}\right)$	$E\left(S_{1,l}I_{1,j}I_{1,k}\right)\;\;\;\;\;$	higher	(15)
$E\left(I_{1,l}S_{1,j}S_{1,k}\right)\approx \;E\left(S_{1,j}S_{1,k}I_{1,l}\right),\;l=1,2,3;j,k=1,2,3,\;j\neq k.$	$pK_{a,3}\left(S_{1,j}S_{1,k}I_{1,l}\right)$	$mr-mE\left(S_{1,j}S_{1,k}\right)$	$E\left(I_{1,l}S_{1,j}S_{1,k}\right)$	lower	(16)
	$pK_{a,3}\left(I_{1,l}S_{1,j}S_{1,k}\right)$	$mr-mE\left(I_{1,l}S_{1,j}\right)$	$E\left(S_{1,j}S_{1,k}I_{1,l}\right)$	higher	(17)

**Table 4 ijms-24-02923-t004:** Cluster prediction based on the linear model proved in Section 2.2.

Compound	pK_a_	Cluster Prediction
C6-OO-, N16-, C17-COO−, C2-OO-	5.29	SSII=SSI=SI=I
C6-OO-, N16-, C17-COO-, C2-OO-, C5-OO-	19.58	SSIIS=SSIS=SSS=SS=S
C6-OO-, N16-, C5-COO-, C2-COO-	5.76	SSSI=SSI=SI=I
C6-COO-, N16-, C5-COO-, C2-COO-, C17-OO-	6.02	SSSII=SSSI=SSI=SI=I
C5-COO-, C6-COO-, C2-COO-, C17-OO-	5.45	SSII=SSI=SI=I
C5-COO-, C6-COO-, C2-COO-, C17-OO-, C15-OO-	5.46	SSIII=SSII=SSI=SI=I
C5-OO-, C6-COO-, C2-COO-, N16-	19.56	SSIS=SSS=SS=S

**Table 5 ijms-24-02923-t005:** Interactions between Bd and the receptor pocket.

Name	Distance (Å)	Category/Type
Cys80:H–Bd:O	3.06	Hydrogen Bond
Leu300:N–Bd:O	2.83	Hydrogen Bond
His110:H–Bd:O	3.15	Hydrogen Bond
Trp111:H–Bd:O	2.56	Hydrogen Bond
Bd:N–Trp111	4.21	Electrostatic/π-Cation
Bd:N–Trp111	4.68	Electrostatic/π-Cation
Trp111–Bd	5.04	Hydrophobic/π-Alkyl
TRP111–Bd	4.60	Hydrophobic/π-Alkyl
Cys298–Bd	5.11	Hydrophobic/Alkyl
Bd–Leu300	4.36	Hydrophobic/Alkyl

**Table 6 ijms-24-02923-t006:** Interactions between the deprotonated Bd (C17C15C2-COOH) and the receptor pocket.

Name	Distance (Å)	Category/Type
His110:H–Bd:O	2.21	Hydrogen Bond
Trp111:N–Bd:O	2.52	Hydrogen Bond
Cys80:H–Bd:O	3.09	Hydrogen Bond
Leu300:H–Bd:N	2.98	Hydrogen Bond
Ala299–Bd:O	3.47	Hydrogen Bond
Thr113:H–Bd:O	2.09	Hydrogen Bond
Bd:N–Trp111	3.71	Electrostatic/π-cation
Bd:O–Trp111	3.89	Electrostatic/π-anion
Cys298–Bd	5.11	Hydrophobic/Alkyl
Trp111–Bd	4.58	Hydrophobic/π-Alkyl
Bd–Val47	5.40	Hydrophobic/π-Alkyl
Bd–Leu300	4.33	Hydrophobic/Alkyl

**Table 7 ijms-24-02923-t007:** Interactions between the deprotonated Bd (C17C15C2-COOH) and the receptor pocket. Complex established by our method (founded on shape theory).

Name	Distance (Å)	Category/Type
Bd:O–His306	4.95	Electrostatic/π-Anion
Bd:O–Trp219	4.17	Electrostatic/π-Anion
Trp20–Bd	3.89	Hydrophobic/π-π T-shaped
Trp111–Bd	3.83	Hydrophobic/π-Alkyl
Trp111–Bd	3.43	Hydrophobic/π-Alkyl
Bd–Cys298	4.16	Hydrophobic/π-Alkyl
Cys303–Bd	4.24	Electrostatic/π-cation
Bd–Leu300	3.89	Hydrophobic/Alkyl
Bd–NDP	5.17	Hydrophobic/π-Alkyl

**Table 8 ijms-24-02923-t008:** Interactions between the deprotonated Bd (C6C15C17-COOH) and the receptor pocket. Complex established by our method (founded on shape theory).

Name	Distance (Å)	Category/Type
Cys80:S–Bd:O	3.48	Hydrogen Bond
Leu300:N–Bd:O	2.85	Hydrogen Bond
Bd:N–His110	3.87	Electrostatic/π-Cation
Bd:O–His306	3.96	Electrostatic/π-Anion
Cys80–Bd	4.96	Hydrophobic/Alkyl
Cys303–Bd	4.57	Hydrophobic/Alkyl
Trp111–Bd	3.67	Hydrophobic/π-Alkyl
Phe115–Bd	5.36	Hydrophobic/π-Alkyl
Bd–Trp111	3.36	Hydrophobic/π-Alkyl
Bd–Val47	4.93	Hydrophobic/π-Alkyl

## Data Availability

The data presented in this study are available in the Appendix A.

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
