# Peer review of "Dissociation Mode of the O–H Bond in Betanidin, pKa-Clusterization Prediction, and Molecular Interactions via Shape Theory and DFT Methods"

_ijms, 2023, doi:10.3390/ijms24032923_

Round 1
Reviewer 1 Report
This is a thorough study of the biological role of betanidin, an antioxidant molecule. I
recommend its publication, but there are two questions I would like the authors to answer in the final version of the paper:
-
How do they deal with conformational flexibility of betanidin? Did they perform a conformational search?
-
Why did the authors choose aldose reductase for their molecular docking studies?
Also, some minor corrections:
-
Electronegativity appears twice in line 105
-
Lines 71-75 seem a bit confusing to me. I guess there are some grammar issues
Author Response
Reponses to Reviewers. Manuscript ID: ijms-1949172
Comments and Suggestions for Authors
REVIEWER 1
Thank you so much for your valuable comments, we are so honored to receive your directions, questions, and suggestions. We expect to answer them according to the profuse revision that you have made.
We have rewritten the paper around a new finding promoted by the comments of the Referees.
The evaluation of the Reviewers has prompted the rewriting of the article including, among others, the following items: 1. Describe the direct method in terms of a linear model that now better explains the pka in terms of energy. 2. With the new model we have been able to elucidate the deprotonation clusters using a combinatorial result. 3. We have made progress in predicting less than half of the calculations of all the routes in each round, an aspect that shows the importance of the new model found. 4. Regarding the discrimination method with Riemannian geometry, we have defined a new distance that involves the Mulliken charges and the atomic masses. With the new measure the geometric optimization is related to the deprotonation chemistry and at the same time the pKa is studied. The previous 4 elements have been written in a new model with their respective proof.
The new model has implied new calculations for more than two weeks. The graphs have been enlarged and the simulated results agree with the theory proved in the new model.
For a complete synchronization with the new implemented linear model proved in section 2.2, we have used our own optimization for the Bd parent, instead the reference parent given in ref [20]. The new Riemann-Mullikan distance has replaced and enriched the old previous report Riemann distance. Thus, the corresponding tables and figures were updated with in new chemical measure involving the atomic mass and the Mulliken charge.
Comment 1
This is a thorough study of the biological role of betanidin, an antioxidant molecule. I recommend its publication, but there are two questions I would like the authors to answer in the final version of the paper:
- How do they deal with conformational flexibility of betanidin? Did they perform a conformational search?
Answer 1
Thank you for the comments made. Below we provide a response to the concern raised.
Line 649 replaced with:
“The computational study starts with a conformational analysis using the software ArgusLab [56] to identify the most stable structure. The optimization of the molecular geometry was performed by using the computational program Gaussian 09 package [57].
In the new manuscript, the reference [30] was changed by the new reference [57]”
Comment 2
- Why did the authors choose aldose reductase for their molecular docking studies?
Answer 2
Thanks a lot. We hope to give the appropriate response to the point you raise.
In the new manuscript we have added the following paragraph:
“This process impacts the NADPH/NADP+ ratio. NADPH is essential for regenerating the reduced form of the intracellular antioxidant glutathione [26]. Factors such as sorbitol accumulation and oxidative stress can complicate diabetes situations, and the inhibition of aldose reductase can prevent these secondary complications [27].
Attending your important comment, we have included new references [26] and [27], strongly related with your question.
In addition, we have updated the bibliographical references.
Comment 3
- Other minor corrections
Electronegativity appears twice in line 105
Answer 3
Thank you. We have removed the repeated addressed word.
Comment 4
- Lines 71-75 seem a bit confusing to me. I guess there are some grammar issues
Answer 4
Thank you so much. We have modified the referred lines as follows:
“The pKa value is an important parameter to elucidate the oxidation mechanism of any substance. Bd belongs to the general group of Betalains. The experimental pKa values of Bd were limited to a single deprotonation step. In this work we have used the Density Functional Theory (DFT) and the density-based solvation model (SMD) to calculate the corresponding pKa values. However, there are several methodologies for computing the pKa, see for example [17], [18], [19].”
Thank you so much again.
The best regards,
The Authors.

Author Response
REVIEWER 2
We appreciate your comments and criticisms; they are an opportunity to improve our work and expose it to you in the best way. For a technical revision of your comments and criticisms we have included new references exclusively written in this letter for supporting the answers. The references will be indexed by parenthesis. The references of the manuscript will be given in brackets, as usual.
We have rewritten the paper around a new finding promoted by the comments of the Referees.
The evaluation of the Reviewers has prompted the rewriting of the article including, among others, the following items: 1. Describe the direct method in terms of a linear model that now better explains the pka in terms of energy. 2. With the new model we have been able to elucidate the deprotonation clusters using a combinatorial result. 3. We have made progress in predicting less than half of the calculations of all the routes in each round, an aspect that shows the importance of the new model found. 4. Regarding the discrimination method with Riemannian geometry, we have defined a new distance that involves the Mulliken charges and the atomic masses. With the new measure the geometric optimization is related to the deprotonation chemistry and at the same time the pKa is studied. The previous 4 elements have been written in a new model with their respective proof.
The new model has implied new calculations for more than two weeks. The graphs have been enlarged and the simulated results agree with the theory proved in the new model.
For a complete synchronization with the new implemented linear model proved in section 2.2, we have used our own optimization for the Bd parent, instead the reference parent given in ref [20]. The new Riemann-Mullikan distance has replaced and enriched the old previous report Riemann distance. Thus, the corresponding tables and figures were updated with in new chemical measure involving the atomic mass and the Mulliken charge.
Paragraph 1 (Comment 1)
The reviewed manuscript concerns betanidin (Bd), the aglycone of betanin, natural pigment with antioxidant activity. Betanidin is a tricarboxylic acid with iminium group, thus, it is easy deprotonated and it can exist in different forms depending on the pH of solution.
Answer
Thanks for your comments. Precisely, because of these properties of Bd and the multiple theoretical and some experimental works, we take this molecule as a reference.
Paragraph 2 (Comment 2)
In the manuscript deprotonation routes of Bd are explored using two methodologies: the direct approach and the approach based on the Riemannian distance.
Answer
We do not present a new methodology for deprotonation based on a pure approach to Riemannian geometry. The paper develops only the common direct approach by pKa, usually approved in Chemistry. Behind each deprotonation route, an exhaustive optimization is performed in expert chemical softwares. The computational study starts with a conformational analysis using the software ArgusLab [56] to identify the most stable structure. The optimization of the molecular geometry was performed by using the computational program Gaussian 09 package [57].
Paragraph 3 (Comment 3)
Deprotonation pathway of Bd using the direct approach has been sensibly studied in ref. 20. New in this manuscript is that the authors considered additional pathways of deprotonations. Finally they obtained a set of pathways; each pathway contains the other set of the calculated pKa values (see Fig. 6). In fact, only the pathway underlined in red, the same as proposed in ref. 20, is reasonable.
Answer
According to the following references, the pathway starting in H-N16 is as possible as the referred red pathway of the comment.
In fact, the same ref.20 highlights:
“The oxidation mechanism of Betanidin (Bd) is highly affected by the medium pH.17,18 Therefore, it is necessary to know the values of pKa and the deprotonation positions of the Bd (Figure 1). In addition, it is generally assumed that the +N16H2 of Bd is deprotonated before the carboxylic acid groups.10.
”
Moreover, in the reference (2) of this letter, we read:
"Surprisingly, our calculations predict that H-N16 proton, not C2-COOH (as it was suggested in some papers: Nilsson 1970; Frank et al. 2005) is most easily ionizable group (Figure 3D and A, respectively)."
It is easy to trace the source of this important assertion about H-N16 and C2-COOH in a number of papers. With the addressed discrepancies related with C17-COOH, H-N16 and C2-COOH, in general, there is not a consensus about the best deprotonation route. There is not a published work that finishes the corresponding research with the referred red route of ref. 20 and prevent of new deprotonation ways with a profuse chemical and/or biological justification.
Moreover in a similar context we also find in [10]: “We have found that each mono-deprotonated form of betanin is a better H donor (lower BDE values) than betanin in its cationic form. In the case of C2-COO- form, the BDE drops by about 10 kcal mol/1 in comparison with cationic form”. In addition, the same reference states: “In a strongly acidic environment, the betanin molecule may exist in a cationized form with an excessive positive charge localized in proximity of the N-1 nitrogen. Because betanin contains three carboxyl groups and potentially ionizable H-N16 and C6-OH protons, in a mild acidic solution it can appear in the form of various zwitterionic states.”
Just as other areas propose other options without moving away from the chemical context, they open the possibility of discovering different routes that are not classically possible. The references are open to all the options of deprotonation routes from a theoretical and/or experimental point of view. We highlight for example the work (1) which supports the N16-H pathway by experiments.
Paragraphs 4 and 5 (Comment 4 and 5)
To propose new pathways the authors used a strictly mathematical approach, which does not take into account the chemical properties of the atoms from which the proton is detached. Using such a wrong approach, the authors obtained absurd results where pKa2 and even pKa3 are lower than pKa1.
The same applies to the second methodology based on the Riemannian distance. Again, this approach is based on math only, all proton breakpoints are treated equally. This is not acceptable for analysis of deprotonation pathways.
Answer
Thanks for your comments.
The paper develops only the direct approach by pKa. The pKa1,2,3 never were referred in the paper as a finding of the Riemannian distance. We did not propose a purely geometrical approach for the exclusive chemical problem of deprotonation. This assertion drops the remaining arguments of the comments. In particular, the syllogism: “Using such a wrong approach, the authors obtained absurd results where pKa2 and even pKa3 are lower than pKa1.” lacks of weight, because the antecedent is false. We insist that the paper only used the direct method based on pKa ; we do not propose a new method for deprotonation routes based on Riemannian geometry.
If the statement “the authors obtained absurd results where pKa2 and even pKa3 are lower than pKa1.” is a chemical assertion, please read (2): In that work is proved that C2-COO-/16N-/C6O is the best deprotonation route. According to our results, the corresponding pKa in three rounds, are the following: First round, pKa of C2-COO=2.78; second round, pKa of C2-COO-/16N= 12.12; third round, pKa of C2-COO-/16N-/C6O = 11.85. This contradicts the proposed law for the Referee: pKa,1 < pKa,2 <pKa,3, for all deprotonations.
In the same paper (2), we found the following routes: The monoanion with the lowest energy corresponds to the N16-H group, followed by the 16N-/C6O- dianion and the C2-COO-/16N-/C6O- trianion (2).
These results verify the assertion of our manuscript about the non-heritage of the preceding deprotonation routes. This ratifies that all the deprotonation routes for each round must be computed.
As it can be checked in every sentence related to the Riemannian geometry, we just use it to complement the calculating of pKa values. The Riemannian analysis is just an invariant method for doing geometrical and algebraical statistics with the emerging data of the chemical optimization procedure of Gaussian and the corresponding pKa. In fact, the paper can be reading without the intervention of the Riemannian theory, and the results of the chemical part of the direct method hold completely. Without the Riemannian analysis, the following results of the direct method hold with great robustness: 1) Only the computation of all possible reasonable deprotonation routes of the inferior cluster can provide the best minima pKa. All the deprotonation routes can be ordered according to the pKa. Following the accepted chemical universal principle about minimal energy, the best deprotonation route in a given round is, without any doubt, the best minima pKa. (See the addressed references [20], (1), (2) and their discrepancies) 2) A local minimum in pKa is not heritable from the preceding round of deprotonation, see again the above answer of comment 3. 3) The values-pKa for the first three rounds are now proved from a chemical principle published in [20,42], that split into two significant statistical groups of routes. Moreover we have proved a linear model for the pKa which can predict the other clusters by using the corresponding energies. See the new section 3.4 and the new Figures 11-19 for predictions. The first round of 6 chemical reasonable routes is divided according to the pKa (without the Riemannian distance) into two groups of 3 routes. The second route of 30 reasonable deprotonation routes is also divided in a superior group of 15 routes of high pka, which is chemically separated (without a Riemannian intervention) from a low pKa with 15 routes. The third round of deprotonation is also split in two groups of 60 routes. Moreover, the cluster prediction of fourth and fifth round were tested with the new model and after long computations, they were verified as we expected. 4) The literature of the direct and the only method of the paper, do not discard any of the permutations for the first, second, and third deprotonations, respectively. Thus, our method is completely realistic, and all the studied deprotonations via pKa are reasonable (See again answer of comment 3). This implies that we have a complete population data of pKa susceptible to be studied with a meaningful mathematical method. For the new manuscript, we included the so called Riemann-Mulliken Distance as a complementary analysis for the following reasons: A) The Riemannian theory considers the geometry of the nucleus (as points), the atomic mass and the Mulliken charges in the molecules, under an invariant perspective (fibers in a quotient space), a notorious fact that absorbs the noisy Euclidean information. The classical Euclidean geometrical approaches usually studied in the optimization chemical softwares do not consider the equivalent classes to avoid unnecessarily repeated computations. The Riemannian theory of the paper has been used in similar chemical problems, such as the determination of equilibrium structures of nanoclusters, see [60]. B) This invariant analysis provides a robust method using the chemical concept of mass and charge for classifying the natural groups divided by the pKa.
This important relationship between the pKa and the geometry optimization intrinsic in the Riemannian-Mulliken distance will be a part of future work.
- C) The Riemannian theory mixed with chemical-physical principles has been successfully applied in other areas (3), (4), (5), (8); in particular, the molecular docking in this paper was performed by a new technique based on Riemannian geometry and Lennard-Jone's potential. The effectiveness of the chemical-geometrical method emerges from a theorem for optimization given in previous work [10].
Another docking method of complementing chemical analysis with advanced geometry and algebraic statistics has also been applied in paper [11]. Recently have appeared several works in high-impact journals completing the physical-chemical analysis with robust mathematical theories, see for example: (3)-(11).
Paragraph 6 (Comment 6)
Permutations of deprotonations that cannot take place lead to results which are not real. It makes no sense to calculate the pKa values for the path that starts from the removal of the C6-OH or N16-H protons. According to the authors results, it even makes no sense to start deprotonation from the C2OOH or C15OOH.
Answer
See again answer of comment 3.
The addressed ref.20 highlights:
“The oxidation mechanism of Betanidin (Bd) is highly affected by the medium pH.17,18 Therefore, it is necessary to know the values of pKa and the deprotonation positions of the Bd (Figure 1). In addition, it is generally assumed that the +N16H2 of Bd is deprotonated before the carboxylic acid groups.10. ”
Moreover, in the reference (2) of this letter, we read:
"Surprisingly, our calculations predict that H-N16 proton, not C2-COOH (as it was suggested in some papers: Nilsson 1970; Frank et al. 2005) is most easily ionizable group (Figure 3D and A, respectively)."
Moreover, experimental studies show the N16-H monoanion of Bd deprotonates before the carboxylate group. This experimental result is obtained from the fluorescence spectra (1). The monoanion with the lowest energy corresponds to the N16-H group, followed by the 16N-/C6O- dianion and the C2-COO-/16N-/C6O- trianion (2).
For the assertion “According to the authors results, it even makes no sense to start deprotonation from the C2OOH or C15OOH.
” please see the first paragraph of answer for comments 4 and 5: We insist that the paper only used the direct method based on pKa , we do not propose a new method for deprotonation routes based on Riemannian geometry.
Paragraph 7 (Comment 7)
In the second part of the manuscript molecular docking of the three forms of Bd in the active site of aldose reductase (AR) followed by MD calculations were performed. It has to be stressed that docking only makes sense when it is justified. The authors write “Returning to anti-radical activity, we consider analyzing the interactions between Bd as the ligand and the enzyme aldose reductase (AR) as the receptor through docking and molecular dynamics.” (page 3, lines 107-109). The problem is that no enzyme is involved in the anti-radical action of Bd. Besides, Bd is not a substrate for AR. As the name of the enzyme suggests, it is involved in the reduction of an aldehyde group, which is not present in Bd. Moreover, Bd as an antioxidant behaves like a reducing agent. AR is also a reducing agent.
Answer
As we have already mentioned, the inhibition of aldose reductase prevents secondary complications in diabetes. This situation is mainly due to the accumulation of sorbitol and oxidative stress associated with these complications. The role of aldose reductase is essential in the polyol pathway, which converts D-glucose to D-sorbitol using NADPH as a cofactor; NADPH is critical to the regeneration of the reduced form of the intracellular antioxidant glutathione [12], [13].
Additionally, in the literature reported in the manuscript [27], [28], [29] and [45], we found recent studies (2017 to 2021) in which natural compounds with antioxidant capacity are tested for possible use as inhibitors of this enzyme since it is known about the toxic potential of synthetic molecules used for the inhibition of AR in patients. We present a computational approach to the possible inhibition of this natural and antioxidant molecule to the AR enzyme.
Finally, in the new manuscript we have included the following paragraph:
This process impacts the NADPH/NADP+ ratio. NADPH is essential for regenerating the reduced form of the intracellular antioxidant glutathione [26]. Factors such as sorbitol accumulation and oxidative stress can complicate diabetes situations, and inhibition of aldose reductase can prevent these secondary complications [27].
In addition, we update the bibliographical references.
Paragraph 8 (Comment 8)
The manuscript is written in a unclear manner. Quite often a given sentence is not related to the previous one. See example (page 3, lines 123-125):
“Two hydroxyl groups (-OH) are connected to the catechol group at the C5 and C6 positions. Another intramolecular hydrogen bond is present at the dihydropyridine ring nitrogen.”
Answer
We modify the paragraph as follows:
Two hydroxyl groups (-OH) are connected to the catechol group at the C5 and C6 positions. Finally, another hydrogen is attached at the nitrogen of dihydropyridine ring.
Paragraph 9 (Comment 9)
There are also quite a lot of minor and major mistakes in the manuscript. Below are three examples:
- Page 4, lines 155-159:
“Therefore, the hydrogens bonded to the oxygens of the carboxyl groups indicate a possible site for a nucleophilic attack. A greater group probability is located in the C15 and C17 positions; on the contrary, in the hydroxyl groups, an attack can be carried out as an electrophilic, with a higher group probability located at position C5.”
This is a conceptual mish-mash.
Answer
We modify the addressed paragraph as follows:
The hydrogens bonded to the oxygens of the carboxyl groups indicate a possible site for a nucleophilic attack, most likely for the group located at the C15 and C17 positions. On the contrary, an electrophilic attack could occur in the hydroxyl groups, with a greater probability for the group found in C5. We notice that an electrophilic attack is less likely to occur than a nucleophilic attack.
- Page 5, lines 175-176:
“The electrons can flow from a region of higher electronegativity to one of lower electronegativity.” This statement is contradictory to the definition of electronegativity.
Answer
We modify the sentence as follows (page 5, lines 181-183):
The calculated electronegativity value indicates that Bd can attract a charge to it. So, in an electron transfer opportunity, the Bd will accept and neutralize electrons from the radicals.
- Page 20, lines 600-601:
“Equation 2 points summarize the SPLET mechanisms (sequential proton-loss electron transfer).” The authors are wrong, equation 2 shows only the dissociation of the phenolic group.
Answer
We modify the paragraph as follows:
The capacity to scavenge free radicals through a hydrogen atom transfer (H•) has been mainly approached through three mechanisms that are often discussed in the literature [13], [55], [57]. We consider the SET (single electron transfer) and the SPLET (sequential proton loss electron transfer) mechanisms.
The SET mechanism consists of one step defined by Equation (1). Here, an electron is transferred to a free radical, forming a radical cation.
The SPLET mechanism consists of two steps (see Equations 2 y 3). The first step refers to deprotonation, and the second is an electron transfer to a radical.
Paragraph 10 (Comment 10)
Finally, I am surprised that this manuscript has been submitted to the molecular biology section. This is another misunderstanding.
Answer
We send this manuscript in response to an editorial invitation and we consider that this research does correspond to the molecular biology section.
References
Note: References 1-13 are included to respond to Reviewer 2 and are not part of the manuscript under review.
- Zabihi, F.; Kiani, F.; Yaghobi, M.; Shahidi, S.A. The Theoretical Calculations and Experimental Measurements of Acid Dissociation Constants and Thermodynamic Properties of Betanin in Aqueous Solutions at Different Temperatures. J Solution Chem 2019, 48, 1438–1460, doi:10.1007/s10953-019-00930-x.
- Gliszczyńska-Świgło, A.; Szymusiak, H.; Malinowska, P. Betanin, the Main Pigment of Red Beet: Molecular Origin of Its Exceptionally High Free Radical-Scavenging Activity. Food additives and contaminants 2006, 23, 1079–1087, doi:10.1080/02652030600986032.
- Arias, E.; Caro-Lopera, F.; Florez, E.; Perez-Torres, J. Two Novel Approaches Based on the Thompson Theory and Shape Analysis for Determination of Equilibrium Structures of Nanoclusters: Cu 8 , Ag 8 and Ag 18 as Study Cases. Journal of Physics: Conference Series 2019, 1247, 012008, doi:10.1088/1742-6596/1247/1/012008.
- Villarreal-Rios, A.L.; Bedoya-Calle, Á.H.; Caro-Lopera, F.J.; Ortiz-Méndez, U.; García-Méndez, M.; Pérez-Ramírez, F.O. Ultrathin Tunable Conducting Oxide Films for Near-IR Applications: An Introduction to Spectroscopy Shape Theory. SN Appl. Sci. 2019, 1, 1553, doi:10.1007/s42452-019-1569-y.
- Quintero, J.H.; Mariño, A.; Šiller, L.; Restrepo-Parra, E.; Caro-Lopera, F.J. Rocking Curves of Gold Nitride Species Prepared by Arc Pulsed - Physical Assisted Plasma Vapor Deposition. Surface and Coatings Technology 2017, 309, 249–257, doi:10.1016/j.surfcoat.2016.11.081.
- Gómez-Urrea, H.A.; Ospina-Medina, M.C.; Correa-Abad, J.D.; Mora-Ramos, M.E.; Caro-Lopera, F.J. Tunable Band Structure in 2D Bravais–Moiré Photonic Crystal Lattices. Optics Communications 2020, 459, 125081, doi:10.1016/j.optcom.2019.125081.
- Gómez-Urrea, H.A.; Bareño-Silva, J.; Caro-Lopera, F.J.; Mora-Ramos, M.E. The Influence of Shape and Orientation of Scatters on the Photonic Band Gap in Two-Dimensional Bravais-Moiré Lattices. Photonics and Nanostructures - Fundamentals and Applications 2020, 42, 100845, doi:10.1016/j.photonics.2020.100845.
- León, A.M.; Velásquez, É.A.; Caro-Lopera, F.; Mejía-López, J. Tuning Magnetic Order in CrI3 Bilayers via Moiré Patterns. Advanced Theory and Simulations 2022, 5, 2100307, doi:10.1002/adts.202100307.
- Ramírez-Velásquez, I.; Bedoya-Calle, Á.H.; Vélez, E.; Caro-Lopera, F.J. Shape Theory Applied to Molecular Docking and Automatic Localization of Ligand-Binding Pockets in Large Proteins. ACS Omega, doi:doi.org/10.1021/acsomega.2c02227.
- Ramirez-Velasquez, I.M.; Velez, E.; Bedoya-Calle, A.; Caro-Lopera, F.J. Mechanism of Antioxidant Activity of Betanin, Betanidin and Respective C15-Epimers via Shape Theory, Molecular Dynamics, Density Functional Theory and Infrared Spectroscopy. Molecules 2022, 27, 2003, doi:10.3390/molecules27062003.
- Valencia, G.M.; Anaya, J.A.; Velásquez, É.A.; Ramo, R.; Caro-Lopera, F.J. About Validation-Comparison of Burned Area Products. Remote Sensing 2020, 12, 3972, doi:10.3390/rs12233972.
- Brownlee, M. The Pathobiology of Diabetic Complications: A Unifying Mechanism. Diabetes 2005, 54, 1615–1625, doi:10.2337/diabetes.54.6.1615.
- Kato, A.; Yasuko, H.; Goto, H.; Hollinshead, J.; Nash, R.J.; Adachi, I. Inhibitory Effect of Rhetsinine Isolated from Evodia Rutaecarpa on Aldose Reductase Activity. Phytomedicine 2009, 16, 258–261, doi:10.1016/j.phymed.2007.04.008.
Thank you so much again.
The best regards,
The Authors.

Round 2
Reviewer 2 Report
I have to repeat again that presented in the manuscript permutations of the all possible places of deprotonation of betanidin (Bd) and calculations of the subsequent pKa values for the each path have nothing to do with reality and are not acceptable. Experimental determination of the real pKa values is based on titration the maximally protonated molecule with base (see scheme below). The pKa is the pH of the solution where the concentration of a given form of HA is equal to the concentration of a form of its conjugated base A‒ (see Henderson-Hasselbalch equation). Therefore, the pKa values of Bd must be ordered in the following way: pKa1 < pKa2 < pKa3 < pKa4 < pKa5 < pKa6. This series must be in accordance with the increasing pH of solution during titration. Consequently, there is only one possible deprotonation path for betanidin (Bd). The idea of this path may vary in details (see Refs 20 and 36 in the manuscript), but it is still the only one path.
|
|
NaOH |
|
NaOH |
|
NaOH |
|
NaOH |
|
NaOH |
|
NaOH |
|
|
Bd+ |
® |
Bd |
® |
Bd1- |
® |
Bd2- |
® |
Bd3- |
® |
Bd4- |
® |
Bd5- |
|
|
pKa1 |
|
pKa2 |
|
pKa3 |
|
pKa4 |
|
pKa5 |
|
pKa6 |
|
In the deprotonation paths proposed by the authors quite often the pKa2 and even the pKa3 are lower than pKa1 (see Figs 6-8 in the revised version and each deprotonation path which starts from the N16-H). Such the result means that during titration with base the pH of solution decreases. Such a ridiculous conclusion is due to the fact that the authors start deprotonation from the places which are hardly deprotonated. According to their own results, the removal of the N16-H or catechol proton first demands the pH of about 10 (see Figs. 7-9 in the revised version). However, in such a basic solution all three carboxyl groups must be deprotonated. The pKa3 4.07 calculated for the carboxylic group (see Fig. 6, red path) means that at pH 4 the half of Bd has two carboxyl groups deprotonated and the other half has three carboxyl groups deprotonated. Undoubtedly, there is no traces of Bd+ at pH 10. Thus, there is no possibility to remove the N16-H or catechol proton before the carboxylic protons.
I would like to add that the authors misinterpret not only their own results but also the results presented by others. To justify the first deprotonation of the N16-H group, the authors refer to a sentence taken out of the context from Ref. 20 (“In addition, it is generally assumed that the +N16H2 of Bd is deprotonated before the carboxylic acid groups.10” in response to review). They ignore the fact that Ref. 20 actually demonstrates that the N16-H proton is deprotonated fifth in line (pKa5). The same is with Ref. 36. Citing it the authors state that “… experimental studies show the N16-H monoanion of Bd deprotonates before the carboxylate group …(1).” It is not true. Firstly, the subsequent pKa values were experimentally determined in Ref. 36 without indicating the places of deprotonation. Secondly, the places of deprotonations were indicated only in theoretical studies presented in this paper. In these studies The N16-H group was not deprotonated at all.
Author Response
We appreciate your comments.
In the attached pdf we respond to your comments.

Round 3
Reviewer 2 Report
The authors should not impute that the reviewer didn’t read the Ref. 10. The Ref. 10 (2006) was not mentioned in the reviews because:
1. There are the deprotonation energies calculated in this paper whereas the authors calculated the pKa values. It is not the same.
2. In Refs 20 (2020) and 36 (2019) it was proven, based on the pKa values, that it is not possible to deprotonate the N16-H proton in the first, second, third or even in the fourth order. The authors hide or ignore these facts.
3. The results obtained by the authors also indicate that it is not possible to deprotonate the N16-H proton in the first, second, or third order. Unfortunately, the authors do not see it.
The second paper recently recommended by the authors to the reviewer just confirms the reviewer’s point.
I only agree with the authors that this discussion has to be closed.
Author Response
Response to Reviewer 2's final comments
We keep the history of all your revisions so that if the article is approved, it is published as this editorial does, and the transparency of our work is shown, and it is published with your revisions so that the scientific community is the one who can evaluate the relevance or not. of your final comments. We appreciate the evaluation and your comments because it allowed us to further refine the article.
The authors